# Neural activity in cortico-basal ganglia circuits of juvenile songbirds encodes performance during goal-directed learning

Jennifer M Achiro[1], John Shen[1], Sarah W Bottjer[2]*

[1]Neuroscience Graduate Program, University of Southern California, Los Angeles, United States; [2]Section of Neurobiology, University of Southern California, Los Angeles, United States

**Abstract** Cortico-basal ganglia circuits are thought to mediate goal-directed learning by a process of outcome evaluation to gradually select appropriate motor actions. We investigated spiking activity in CORE and SHELL subregions of the cortical nucleus LMAN during development as juvenile zebra finches are actively engaged in evaluating feedback of self-generated behavior in relation to their memorized tutor song (the goal). Spiking patterns of single neurons in both CORE and SHELL subregions during singing correlated with acoustic similarity to tutor syllables, suggesting a process of outcome evaluation. Both CORE and SHELL neurons encoded tutor similarity via either increases or decreases in firing rate, although only SHELL neurons showed a significant association at the population level. Tutor similarity predicted firing rates most strongly during early stages of learning, and SHELL but not CORE neurons showed decreases in response variability across development, suggesting that the activity of SHELL neurons reflects the progression of learning.
DOI: https://doi.org/10.7554/eLife.26973.001

*For correspondence: sarahbottjer@gmail.com

**Competing interests:** The authors declare that no competing interests exist.

## Introduction

Recurrent cortico-basal ganglia circuits mediate procedural skill learning, which involves goal-oriented evaluation of behavioral outcomes to gradually select appropriate motor actions. Vocal learning in songbirds provides a powerful model for studying the control of experience-dependent skill learning by cortico-basal ganglia circuits during development. Similar to infants learning speech, juvenile songbirds memorize vocal sounds of an adult tutor. They then progressively refine their own vocal behavior to imitate the tutor song through iterative comparisons between feedback of their own vocalizations and tutor sounds. Successful acquisition requires the evaluation of behavioral feedback against a representation of the goal (tutor song) to guide the gradual acquisition of an accurate imitation during the sensorimotor stage of vocal learning.

Neural control of vocal learning in juvenile zebra finches (*Taeniopygia guttata*) is vested in cortico-basal ganglia loops that emanate from the cortical nucleus LMAN (lateral magnocellular nucleus of the anterior nidopallium; *Figure 1*) (*Aronov et al., 2008*; *Bottjer et al., 1984*; *Scharff and Nottebohm, 1991*). CORE and surrounding SHELL subregions of LMAN make parallel connections through the basal ganglia and thalamus (*Bottjer, 2004*; *Luo et al., 2001*; *Johnson et al., 1995*; *Gale et al., 2008*; *Person et al., 2008*; *Iyengar et al., 1999*; *Paterson and Bottjer, 2017*) and appear functionally similar to sensorimotor and associative cortico-basal ganglia loops that contribute to different aspects of motor learning in mammals (*Thorn et al., 2010*; *Yin et al., 2008*; *Samejima and Doya, 2007*; *Ashby et al., 2010*; *Redgrave et al., 2010*; *Graybiel, 2008*; *Gremel and Costa, 2013*; *Yin et al., 2009*; *Kupferschmidt et al., 2017*; *Alexander and Crutcher, 1990*). The CORE pathway

mediates vocal motor production in juvenile songbirds (*Elliott et al., 2014*; *Scharff and Nottebohm, 1991*; *Aronov et al., 2008*), whereas the SHELL pathway is involved in evaluating sensorimotor learning; lesions in the SHELL pathway of juvenile birds impair the ability to copy tutor song syllables, but do not cause motor disruption of song (*Bottjer and Altenau, 2010*) (*Figure 1—figure supplement 1*). This disruption of learning, but not motor performance, supports the idea that SHELL circuitry helps to evaluate whether self-generated vocalizations match learned tutor sounds.

Within the SHELL subregion of LMAN in juvenile birds, distinct subpopulations of neurons respond selectively to playback of either their learned tutor song or their own self-generated song (*Achiro and Bottjer, 2013*), indicating that the SHELL pathway has access to neural representations of the goal behavior and the current version of the bird's own song (*Figure 1—figure supplement 2*).

In contrast, CORE neurons are not selective for different song types at the onset of sensorimotor learning, but gradually become selectively tuned to playback of their own song (*Doupe, 1997*; *Solis and Doupe, 1997*). In addition, CORE neurons that project to vocal motor cortex (RA, *Figure 1*) send transient axon collaterals into the SHELL pathway, such that a copy of the motor signal is conveyed into the SHELL circuit only in juvenile birds (*Miller-Sims and Bottjer, 2012*) (*Figure 1—figure supplement 3*). Remarkably, the large population of tutor-tuned neurons in LMAN-SHELL is gone by late stages of sensorimotor learning, suggesting a key role for this subpopulation during a restricted period of development. These developmental changes suggest that the SHELL pathway may evaluate feedback about current vocal performance in relation to a memory of the tutor song during early sensorimotor learning and transmit that evaluation to CORE and other motor pathways (*Achiro and Bottjer, 2013*; *Bottjer et al., 2010*). We evaluated this idea by testing the activity of CORE and SHELL neurons in singing juvenile birds as they are actively involved in sensorimotor learning. The results support the idea that CORE neurons participate in motor-related actions and that activity in both CORE and SHELL neurons reflects evaluation of feedback of motor performance against the goal of memorized tutor sounds.

## Results

### Neurons in both CORE and SHELL subregions of LMAN exhibit singing-related neural activity in juvenile birds

We recorded single neurons in CORE and SHELL subregions of LMAN in juvenile zebra finches that had completed memorization of a tutor song and begun to practice their song vocalizations (43–60 dph). The majority of neurons in both CORE and SHELL showed significant modulation of firing rate during singing (*Table 1*; see Materials and methods). Among neurons that exhibited a significant change in firing rate during song production, approximately 65–70% showed excitation in both CORE and SHELL whereas the remainder were suppressed. Excitatory response strength was marginally higher in SHELL neurons, whereas suppressed response strength did not differ between CORE and SHELL (*Table 1*; Mann-Whitney tests: singing-excited neurons U = 1768, p=0.06, singing-suppressed neurons U = 459, p=0.90). *Figure 2* shows spiking activity of example CORE and SHELL neurons that exhibited average levels of excitation (top) or suppression (bottom) during singing. Singing-excited neurons in both CORE and SHELL showed a significant increase in bursting during singing (fraction of spikes with interspike intervals <10 ms; *Figure 2—figure supplement 1*) as shown previously for LMAN-CORE neurons in juvenile birds (*Olveczky et al., 2005*).

Firing rates of neurons in CORE and SHELL were also similar during production of different syllable types and lacked temporal specificity across renditions of the same syllable type (*Figure 2—figure supplement 2*). Thus, spiking was relatively sparse and variable within and across syllable types in both CORE and SHELL neurons, as reported previously in CORE for both juvenile and adult birds during playback and singing (*Doupe, 1997*; *Doupe and Solis, 1997*; *Kao et al., 2008*; *Olveczky et al., 2005*; *Solis and Doupe, 1997*). In summary, the basic profile of neural activity during song production was similar between SHELL and CORE neurons.

Despite these overall similarities in neural activity between CORE and SHELL, the population of SHELL neurons would not be expected to show a coordinated increase in firing rate prior to the onset of singing if the SHELL pathway lacks a role in vocal motor production (*Bottjer and Altenau, 2010*). In accord with this idea, population histograms of mean-subtracted responses in pre-singing excited neurons showed synchronous increases in firing rates prior to syllable onsets in CORE neurons,

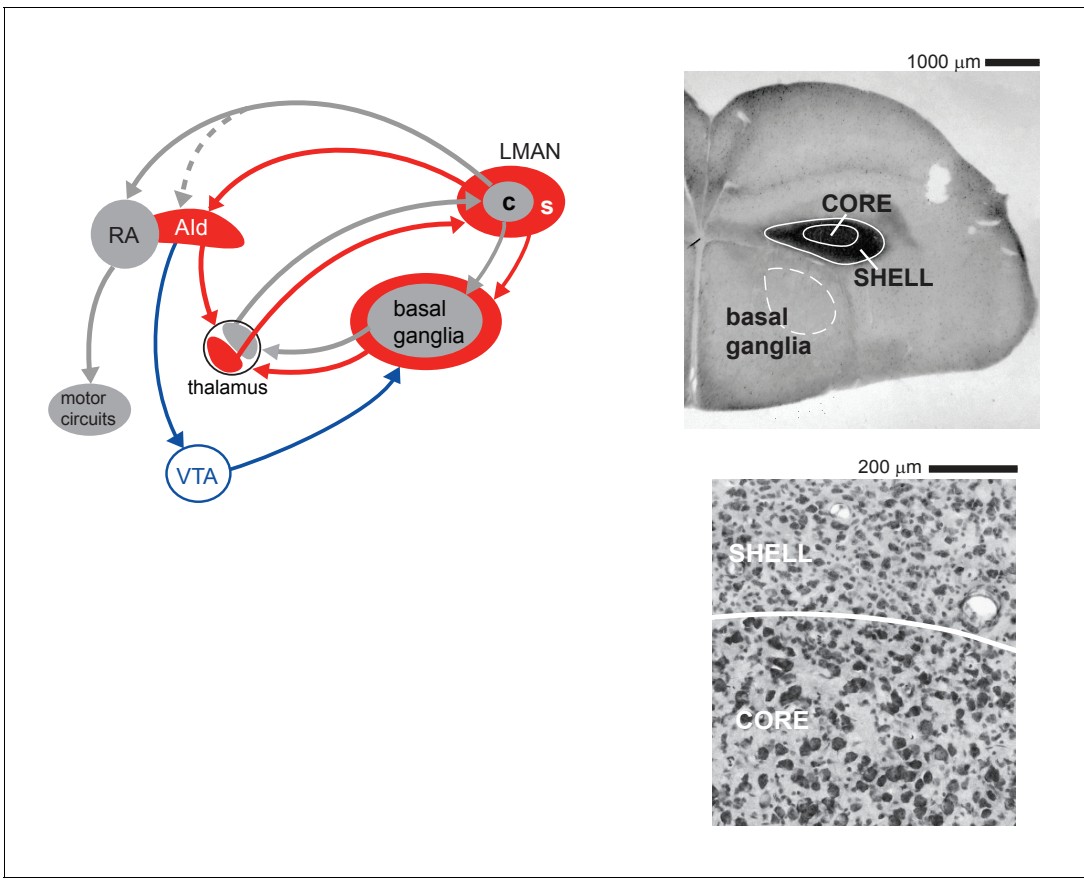

**Figure 1.** Cortico-basal ganglia circuits for vocal learning in juvenile zebra finches. Left: CORE (c, gray) and SHELL (s, red) subregions of the cortical nucleus LMAN give rise to parallel recurrent loops through the basal ganglia. LMAN-CORE projects to vocal motor cortex (RA); this pathway drives vocal motor output in juvenile birds. LMAN-SHELL projects to a region of motor cortex that is adjoined to the lateral margin of RA (AId); this pathway does not drive motor output but is involved in learning. SHELL also forms a trans-cortical loop via AId that converges with the CORE-SHELL basal ganglia loops in a dorsal thalamic zone. A transient projection from CORE to AId (dashed line) is present only in juvenile birds and creates a site of integration between CORE and SHELL pathways in AId during the learning period. The dorsal thalamus feeds back to LMAN and feeds forward to the premotor cortical area, HVC (High Vocal Center), via medial MAN (latter pathway not shown for clarity). Right: upper panel shows a low-power coronal section containing CORE and SHELL regions of LMAN as well as the anterior basal ganglia (Area X, dashed white outline, is an anatomical subregion of basal ganglia in songbirds; it contains both striatal and pallidal cells and is necessary for vocal learning). Calbindin expression (dark staining) demarcates both CORE and SHELL regions by labeling terminals of afferent thalamic axons. Lower panel shows a high-magnification Nissl-stained coronal view of the border between CORE and SHELL subregions, which are distinguished by the higher density of magnocellular neurons within CORE.

DOI: https://doi.org/10.7554/eLife.26973.002

The following figure supplements are available for figure 1:

**Figure supplement 1.** Lesions of AId prevent vocal learning in juvenile birds.
DOI: https://doi.org/10.7554/eLife.26973.003

**Figure supplement 2.** Distinct populations of neurons in LMAN-SHELL respond to either tutor song or self-generated song (own song).
DOI: https://doi.org/10.7554/eLife.26973.004

**Figure supplement 3.** Individual LMAN-CORE neurons send axon collaterals into both RA and AId only in juvenile birds.
DOI: https://doi.org/10.7554/eLife.26973.005

whereas SHELL neurons revealed no evidence for coordinated pre-singing increases in firing rate (*Figure 3*). The lack of time-locked premotor activity in SHELL neurons is consistent with the absence of motor abnormalities following lesions that disrupt SHELL circuitry (*Bottjer and Altenau, 2010*), indicating a non-motor role in learning.

**Table 1.** Response strength during episodes of singing.
Standardized response strength (mean ± s.e.m.) for CORE and SHELL neurons in LMAN that showed significant excitation or suppression during song production compared with quiet baseline periods (see Materials and methods).

| | CORE | | SHELL | |
|---|---|---|---|---|
| | Fraction | Response strength | Fraction | Response strength |
| Excited | 0.72 (66/92) | 7.06 ± 0.71 | 0.65 (66/102) | 7.28 ± 0.44 |
| Suppressed | 0.28 (26/92) | −7.32 ± 1.15 | 0.35 (36/102) | −5.82 ± 0.45 |

DOI: https://doi.org/10.7554/eLife.26973.009

## Neural activity in LMAN reflects similarity of self-generated syllables to tutor syllables

A large proportion of SHELL neurons are specifically tuned to playback of either the tutor song or the current version of self-generated song during early stages of sensorimotor integration when CORE neurons drive vocal motor output and lesions of LMAN pathways cause disruption of song learning (*Achiro and Bottjer, 2013*; *Aronov et al., 2008*; *Bottjer et al., 1984*; *Olveczky et al., 2005*; *Scharff and Nottebohm, 1991*; *Bottjer and Altenau, 2010*). This pattern suggests that LMAN is involved in sensorimotor learning in juveniles, but direct support is lacking for the idea that the activity of individual LMAN neurons represents how well syllables match the tutor song during singing. We tested this idea in juvenile birds (43–60 dph) by comparing neural activity during singing with the acoustic similarity of self-generated syllables to tutor syllables (see Materials and methods): regressions of baseline-corrected firing rates against tutor similarity were performed for each neuron. Unexpectedly, this analysis revealed that firing rates of cells in both CORE and SHELL could either increase or decrease as a function of tutor similarity: approximately half of all neurons in each subregion showed increased firing rates for syllables with higher tutor similarity (positive slopes, r values > 0), whereas the other half showed increased firing rates for syllables with lower tutor similarity (negative slopes, r values < 0) (*Table 2*). *Figure 4A* shows examples of syllable utterances with relatively high or low acoustic similarity to the closest tutor syllable, along with the corresponding firing rate of a SHELL neuron that showed a negative correlation with tutor similarity (*Figure 4B*). This neuron showed a low firing rate during production of syllables with high tutor similarity and excitation during production of syllables with low tutor similarity. Thus, LMAN neurons can encode similarity between self-generated utterances and tutor song via either increases or decreases in firing rate.

To estimate the fraction of single neurons showing a significant relationship between firing rate and tutor similarity, we performed repeated permutation tests by generating 1000 random shuffles of the relationship between firing rate and tutor similarity for each cell (*O'Connor et al., 2010*); see Materials and methods). The percent of significant neurons compared to these random distributions was 5.5% in CORE and 10.8% in SHELL (*Table 2*) (chi-square test between CORE and SHELL proportions = 1.98, p=0.18). CORE and SHELL neurons were evenly split between positive and negative associations of firing rate to degree of tutor song matching, and had fairly comparable r values (averages ranging from +0.22 to −0.31). In summary, tutor similarity predicted the firing rates of a relatively small percentage of single neurons in both CORE and SHELL, with somewhat more neurons in SHELL showing a significant correlation.

To test whether similarity to tutor syllables modulates neural activity at the population level, we analyzed the regressions of baseline-corrected firing rates with tutor similarity across all neurons using a mixed-effects linear regression model (fixed and random effects for tutor similarity nested within a random intercept for neurons; see Materials and methods). This analysis yielded a significant effect for SHELL neurons (t = −2.23, p=0.035) but not for CORE neurons (t = 0.91, p=0.37), indicating that the population activity of SHELL neurons during singing reflected tutor-matching performance. Because positive or negative relationships between firing rate and tutor similarity may reflect unique processes of evaluating tutor similarity (*Table 2*), we examined these two categories separately. Using the mixed-effects linear regression model to provide a descriptive assessment of the magnitude of positive and negative associations, we found significant effects for positive and negative relationships between firing rate and tutor similarity in both CORE and SHELL neurons (CORE positive: t =

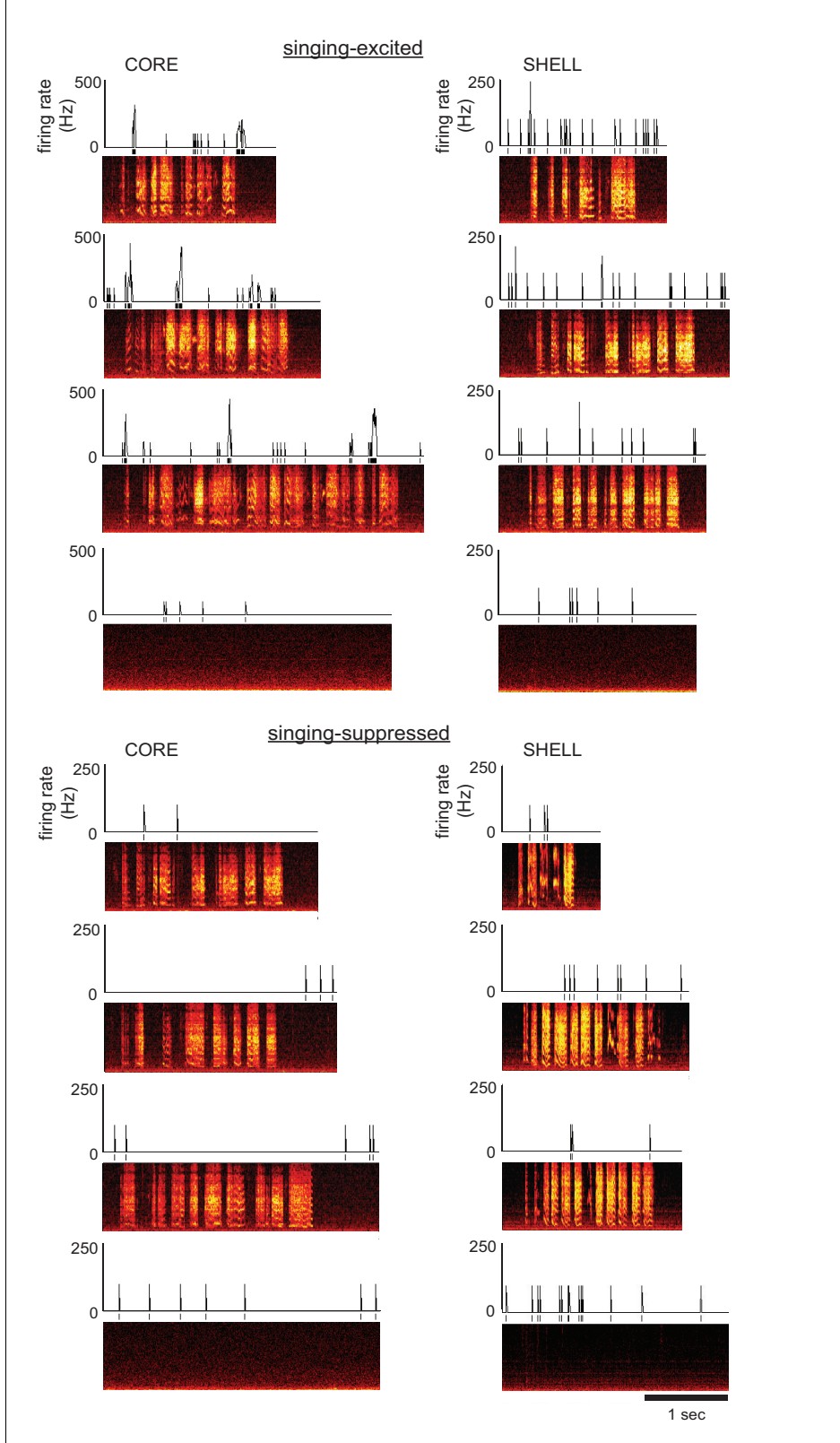

**Figure 2.** Single neurons in both CORE and SHELL subregions of LMAN showed singing-related activity in juvenile songbirds. Left: Examples of two different CORE neurons during singing in juvenile birds showing either excitation (top; 54 dph) or suppression (bottom; 43 dph) compared with quiet baseline periods. Spectrograms depict three example singing episodes (frequency, 0–8 kHz, over time) and one non-singing baseline period; time-aligned

*Figure 2 continued on next page*

*Figure 2 continued*

spikes and corresponding firing rates (spikes/s; 10 ms bin size) are shown above. Right: Examples of two different SHELL neurons during singing in juvenile birds showing either excitation (top; 43 dph) or suppression (bottom; 50 dph), as in left panels.

DOI: https://doi.org/10.7554/eLife.26973.006

The following figure supplements are available for figure 2:

**Figure supplement 1.** Spike bursts increased in excited but not suppressed neurons during singing.
DOI: https://doi.org/10.7554/eLife.26973.007
**Figure supplement 2.** LMAN neurons had low selectivity for different syllable types.
DOI: https://doi.org/10.7554/eLife.26973.008

---

3.42, p=0.003; negative: t = −2.90, p=0.004; SHELL positive: t = 4.02, p=0.001; negative: t = −2.80, p=0.009). To illustrate these relationships, *Figure 5* shows response strengths during production of syllables with low versus high tutor similarity (bottom versus top 50% ranked by tutor similarity; Materials and methods) separated by whether cells showed positive (left panels) or negative (right panels) slope values. These data indicate that the majority of neurons in both CORE and SHELL showed either higher or lower firing rates during production of syllables with higher tutor similarity.

An alternative interpretation of the significant association between firing rate and tutor similarity in SHELL is that firing rates were modulated by more prototypical juvenile utterances, that is, those closer to the center of the distribution of acoustic features for a given syllable type (see Materials and methods). We tested whether the significant correlation to tutor similarity we observed in SHELL neurons reflected a tendency to encode highly prototypical syllables using the mixed-effects regression model described above. This analysis showed a non-significant effect (t = −1.53, p=0.136), and there was no relationship between indices of tutor similarity and prototypicality (data not shown), indicating that tutor-similar syllable renditions are not more prototypical. Thus, the relationship between firing rates and tutor similarity in SHELL neurons does not appear to be based on prototypicality of self-generated syllables.

## Variability in neural responses of both CORE and SHELL neurons reflects similarity of self-generated syllables to tutor syllables

To test if degree of similarity to tutor song also modulates firing rate variability of LMAN neurons in 43–60 dph birds, we calculated the CV (coefficient of variation) of firing rate during production of syllable renditions with low versus high similarity to tutor syllables (bottom versus top 50% of tutor similarity scores). Both CORE and SHELL neurons showed increased firing rate variability during production of syllable utterances with low tutor similarity compared to those with high similarity (*Figure 6A*; Wilcoxon signed-rank tests, CORE Z = −5.70, p<0.001; SHELL Z = −4.48, p<0.001). Thus, variability of firing rates during singing reflected the degree of tutor song matching in both CORE and SHELL neurons such that responses were less variable for syllable renditions that were more similar to tutor song.

One possible caveat is that the decreased variability of firing rate for utterances with higher tutor similarity might simply reflect a smaller number of syllable types in this category, whereas low-similarity utterances might include many different syllable types. To test this idea, we assessed the fraction of syllable types (across all syllable renditions) that were produced within the high- versus low-similarity categories for each day of singing and found no significant difference (*Figure 6B* left; Mann-Whitney U = 693, p=0.06). In addition, there was no difference in the scatter of syllable renditions within a syllable type between the high- and low-similarity to tutor categories (*Figure 6B* right; Mann-Whitney U = 867.5, p=0.90). In summary, these data indicate that variability in firing rate was higher across the populations of both CORE and SHELL neurons during production of syllable renditions with low tutor similarity compared to production of syllable renditions with high tutor similarity.

## Behavioral and neural changes during song development
### Behavioral expression of learning
The developmental span over which we recorded birds' vocal behavior (43–60 dph) corresponds with a transition from highly variable 'subsong' to 'plastic song' with less behavioral variability and a

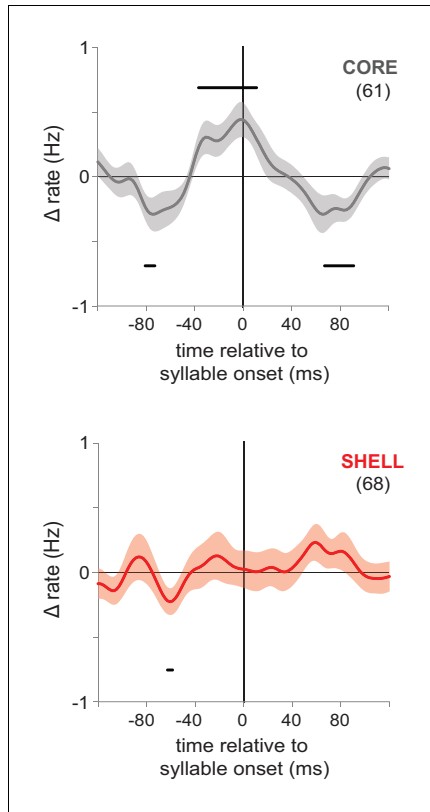

**Figure 3.** Pre-singing activity aligned to syllable onsets showed coordinated premotor activity in CORE but not SHELL. Pre-singing activity in CORE (gray, top panel) and SHELL (red, bottom panel) in juvenile birds for all neurons that showed significant excitation prior to syllable onsets. Solid lines show smoothed mean-subtracted population rate histograms aligned to syllable onsets (syllable onset at time 0), shading indicates s.e.m. Bars above and below the traces indicate times at which the rate change is significant (95% confidence interval outside of zero); n's are indicated in parentheses. Population histograms aligned to syllable offsets showed no significant changes for either CORE or SHELL offset-excited neurons (data not shown).

DOI: https://doi.org/10.7554/eLife.26973.010

The following source data is available for figure 3:

**Source data 1.** Pre-singing spiking activity of individual CORE and SHELL neurons.

DOI: https://doi.org/10.7554/eLife.26973.011

corresponding increase in similarity to tutor song. If the modulation of neural activity by tutor similarity we observed above reflects some aspect of learning, then it should change during the progression of learning. To test this idea, we assessed the degree of song development for each day of singing for each bird by calculating goodness-of-fit coefficients from an exponential fit to the distribution of syllable durations (see Materials and methods; *Figure 7—figure supplement 1*). Juveniles in subsong (but not plastic song) produce a graded distribution of syllable durations, which is well-fit by an exponential function (lower numbers indicate better fit; (*Aronov et al., 2011*; *Tchernichovski et al., 2004*; *Aronov et al., 2008*; *Johnson et al., 2002*). Comparing degree of song development with acoustic similarity to tutor across average syllable types yielded a significant correlation (*Figure 7A*; r = 0.23, p=0.03), supporting the canonical understanding that juvenile birds produce syllable renditions that are more similar to tutor syllables as sensorimotor integration progresses.

We then examined individual syllable utterances with high versus low tutor similarity within each day of singing. Surprisingly, we found that birds produced many utterances with relatively high tutor similarity during subsong (*Figure 7B*, right). Furthermore, the level of tutor matching among these utterances was fairly constant across development such that there was no significant correlation between tutor similarity and song development (r = 0.13, p=0.42). This striking finding indicates that some of the highly variable vocalizations produced by birds during early sensorimotor learning are good matches to tutor syllables. Birds in subsong also produced many low-similarity utterances (*Figure 7B*, left), and the similarity between these poorly-matched renditions and tutor syllables increased as song development progressed, reflecting the expected increase in tutor matching (r = 0.34, p=0.03; cf. *Figure 7A*). Thus, birds in subsong produce some utterances with tutor similarity as high as that of birds in later stages of song development, and renditions with lower tutor similarity at the beginning of sensorimotor integration are gradually eliminated.

These data raise the possibility that syllable renditions with high versus low tutor similarity (*Figure 7B*) were distinct in other ways. For example, high-similarity syllable renditions in subsong birds might consist of simple harmonic stacks, whereas low-similarity renditions might consist of syllable types with complex modulations. We examined whether individual syllable types were selectively represented in high- versus low-similarity categories for each day of singing, and found that the syllable type that was produced most frequently in the high-similarity category (highest percentage of renditions among all syllable types) was also produced most frequently in the low-similarity

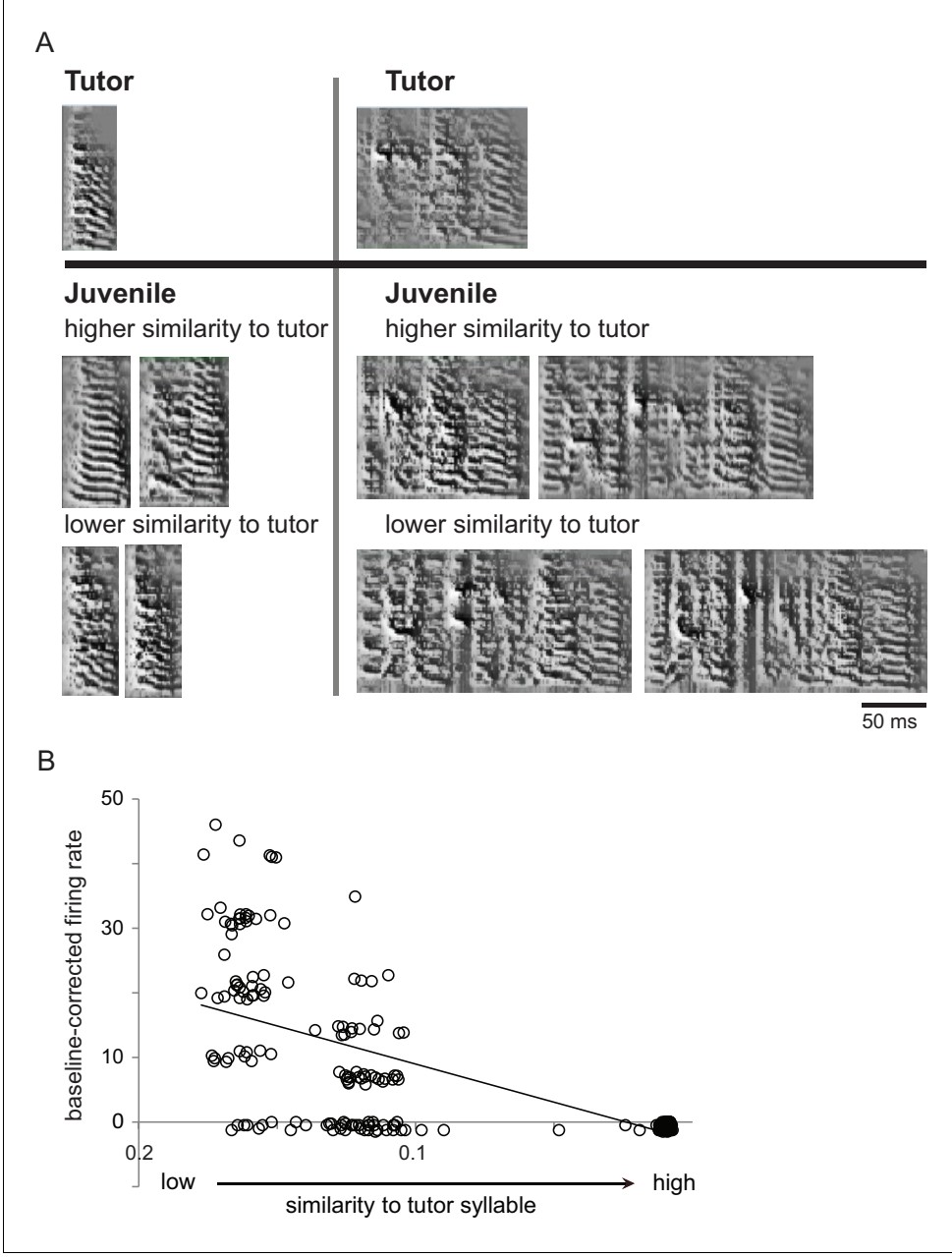

**Figure 4.** Single LMAN neurons encoded similarity to tutor song in singing juvenile birds. (**A**) Examples of juvenile syllables with relatively high or low similarity to tutor syllables. Spectrograms (frequency, 0–8 kHz, over time) showing two different tutor syllables (top) and examples of juvenile renditions from a 59 dph juvenile bird (bottom) which were matched to the corresponding tutor syllable and showed either relatively high or low similarity to that tutor syllable. (**B**) Baseline-corrected firing rates across all renditions of all syllable types for a SHELL neuron from the same bird during the same period of singing as in A; this cell showed a significant negative correlation between baseline-corrected firing rates and tutor similarity of self-generated utterances.
DOI: https://doi.org/10.7554/eLife.26973.012

category. The most common syllable type produced was the same for both high- and low-similarity renditions in 57% of vocal recording sessions across all birds, indicating that a single syllable type did not typically predominate in either low- or high- similarity categories. This pattern is consistent with the finding above that syllable renditions within both high- and low-similarity categories represent multiple syllable types (**Figure 6B**). In addition, qualitative examination revealed that syllable

**Table 2.** Tutor similarity modulates baseline-corrected firing rates in single LMAN neurons.
Single neurons showed either positive or negative slopes for the regression of firing rate against tutor similarity. The incidence of neurons across the population that had either positive (increased firing rates for syllables with higher tutor similarity) or negative (increased firing rates for syllables with lower tutor similarity) r values was similar for CORE and SHELL. Most single neurons had nonsignificant r values, but clear effects were observed at the population level (see **Figure 5**, text).

| | Positive slope (r > 0) | Negative slope (r < 0) | |
|---|---|---|---|
| CORE | 48 | 50 | Total cell number (n = 98) |
| | 49.0 | 51.0 | Percent |
| | 0.082 | −0.065 | Mean of r value across all cells |
| | 3.3 | 2.2 | **Estimated % significant CORE cells = 5.5** |
| | 0.22 | −0.21 | Approximate mean r value for significant cells (n = 5) |
| | | | |
| SHELL | 62 | 60 | Total cell number (n = 122) |
| | 50.8 | 49.2 | Percent |
| | 0.062 | −0.081 | Mean of r value across all cells |
| | 4.5 | 6.3 | **Estimated % significant SHELL cells = 10.8** |
| | 0.21 | −0.31 | Approximate mean r value for significant cells (n = 13) |

DOI: https://doi.org/10.7554/eLife.26973.013

types with the most renditions in high- and low-similarity categories included both simple and complex syllables, although we noted a slight tendency for low-similarity renditions to include more frequency modulation. Thus, syllable renditions with high versus low tutor similarity did not show a strong pattern of syllable type.

## Neural representation of performance during learning

How did neural activity in LMAN encode these changes in tutor similarity as learning progressed? Absolute values of response strength across song development increased in both CORE and SHELL neurons regardless of whether syllable renditions had high or low similarity to tutor syllables (data not shown). This pattern agrees with previously published work showing increases in firing rate of RA neurons as song learning progresses (*Ölveczky et al., 2011*), suggesting generic increases in firing rate with development. However, LMAN circuitry is critical for early sensorimotor learning when the number of tutor-tuned SHELL neurons is high (*Achiro and Bottjer, 2013*), indicating that the involvement of LMAN may be high initially when the goal representation is strong and decrease as learning progresses. In accord with this idea, correlations between firing rate and tutor similarity across development revealed that the activity of LMAN neurons tracked vocal performance during sensorimotor learning (*Figure 8*). Both CORE and SHELL neurons showed stronger associations between firing rate and tutor similarity during early stages of song development, and weaker associations as sensorimotor learning progressed. These trends were significant in CORE and SHELL neurons with positive associations and in CORE neurons with negative associations; SHELL neurons with negative slopes showed a weak but non-significant association between firing rate and tutor similarity across development. However, when the two bottom outliers in the panel for SHELL-negative slopes were removed the association was significant, r = 0.35, p=0.006. This pattern indicates that the activity of both CORE and SHELL neurons reflects the degree of song learning: tutor similarity is a better predictor of firing rate during early sensorimotor learning when many utterances have low tutor similarity (*Figure 7B*), and the strength of this association decreases in parallel with developmental increases in motor performance as song learning progresses.

The variability of firing rate paralleled the behavioral changes in tutor similarity across development (*Figure 7B*) in SHELL but not CORE neurons. CV of firing rate in SHELL neurons decreased over the course of song development only during production of syllable renditions that had low similarity to tutor during early sensorimotor learning (*Figure 9*; low similarity, r = −0.24,p=0.01; high similarity, r = −0.08, p=0.42). Thus, variability of firing in SHELL neurons did not change for utterances with high similarity to tutor song across development. In contrast, the CV of firing rate in CORE neurons did not

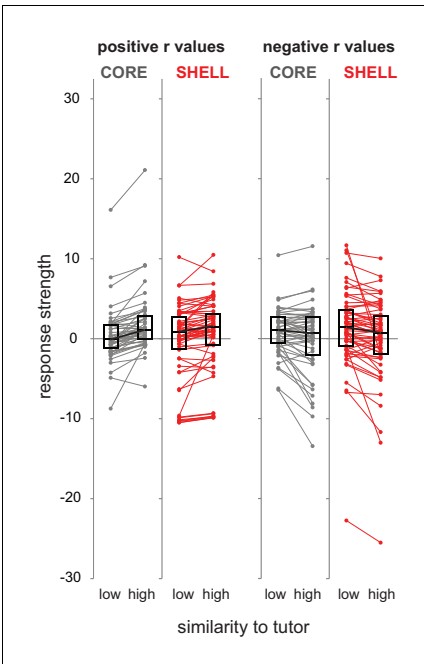

**Figure 5.** Similarity to tutor syllables modulated firing rate in either a positive or negative direction across the population of LMAN neurons. Standardized response strength for each neuron in CORE (gray) and SHELL (red) during production of syllable renditions representing low versus high similarity to corresponding tutor syllables based on a median split of tutor similarity scores (bottom versus top 50% of tutor similarity scores). Left panels, neurons that showed positive slopes in the regression analysis (r values > 0); right panels, neurons that showed negative slopes in the regression analysis (r values < 0) (left panels: CORE n = 48, SHELL n = 62; right panels: CORE n = 50, SHELL n = 60; two outliers have been removed from CORE for clarity of exposition). Box plots represent median and first/third quartiles.

DOI: https://doi.org/10.7554/eLife.26973.014

change as a function of song development regardless of whether syllable renditions had high or low tutor similarity (*Figure 9*; low similarity, r = 0.04, p=0.73; high similarity, r = 0.05, p=0.61). These data show that the variability of firing in SHELL neurons reflected the progression of learning, with higher neural variability early in development for renditions with lower similarity to tutor song, and lower neural variability later in development as the incidence of poorly-matched syllable renditions decreased.

## Discussion

We tested whether spiking activity in LMAN of juvenile birds reflects the acoustic similarity of self-generated vocal utterances to memorized tutor syllables. Neurons in both CORE and SHELL subregions of LMAN showed singing-related activity that varied as a function of tutor song similarity during early stages of sensorimotor integration. This pattern represents the first discovery of an online correlate of song performance during learning in juvenile birds and suggests LMAN as a key component of circuitry that evaluates current behavior in relation to a goal behavior during procedural learning. Although neural activity across the population in both CORE and SHELL was modulated by degree of similarity between self-generated sounds and memorized tutor sounds, this trend was stronger across the population of SHELL neurons. As in prior work, we found that spiking activity in CORE neurons supports the idea that CORE drives vocal motor output in juvenile birds (*Aronov et al., 2008*). In contrast, neural activity in SHELL neurons did not exhibit coordinated premotor increases in firing rate, which is consistent with the lack of projections from SHELL to downstream motor circuitry and the absence of motor disruption following lesions to the SHELL pathway (*Bottjer and Altenau, 2010*; *Bottjer et al., 2000*) (*Figure 1*).

One interpretation of this pattern of results is that SHELL circuitry acts primarily as a 'critic' to evaluate comparisons between self-generated and tutor sounds, while CORE circuitry serves as an 'actor' that drives vocal motor output and receives instruction from the critic in order to bias action selection over the course of learning (*Barto et al., 1983*; *Graybiel, 2008*). The idea of an instructive function by 'critic' circuitry is complicated (here as elsewhere), since the recurrent loop architecture of cortico-basal ganglia circuits means that error or reinforcement signals (which are themselves widely distributed) may be propagated into multiple pathways (*Lau et al., 2017*). Thus in the current results it is difficult to know whether evaluative signals may originate in SHELL and be transmitted to CORE in order to instruct future motor actions. We favor this idea as a working model; the current data serve as a foundation for tests of this idea in future studies.

### Parallels between behavioral and neural indices of learning

Interestingly, some individual syllable renditions were similar to tutor syllables even in subsong birds during early sensorimotor integration, indicating a surprising capacity to produce relatively mature

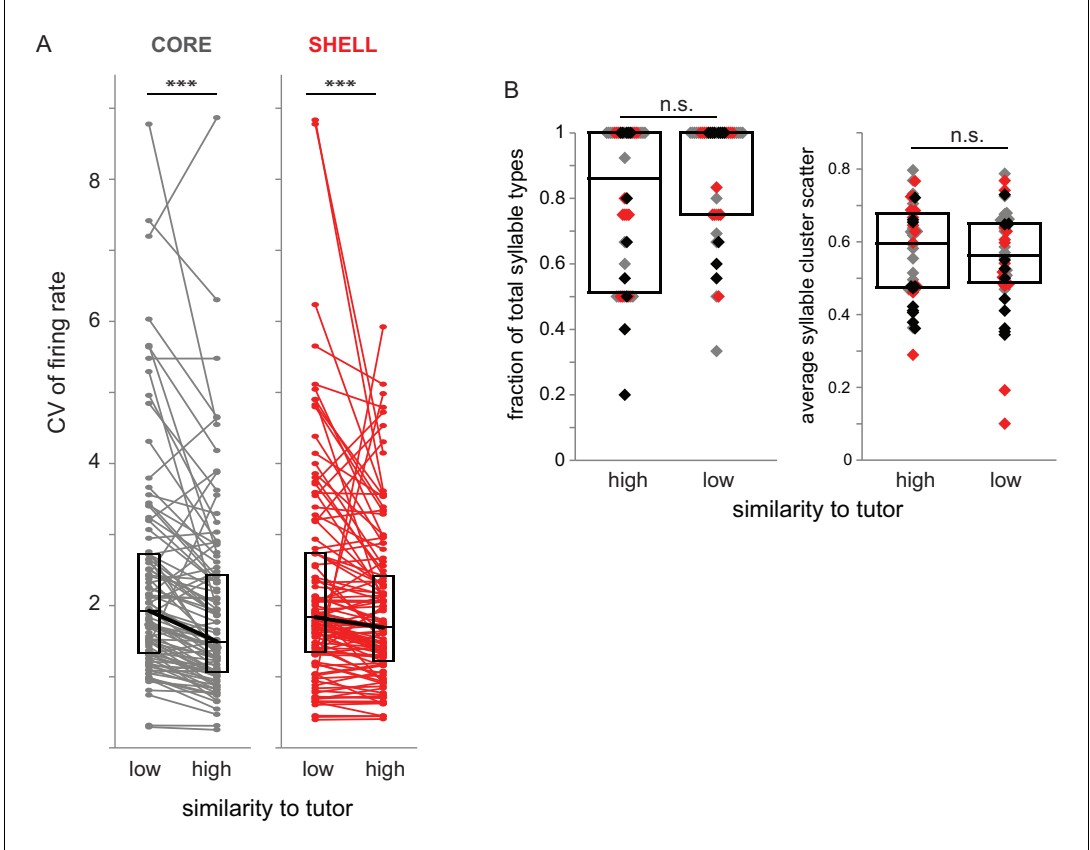

**Figure 6.** Variability of firing rate was higher during syllable renditions with low tutor similarity in both CORE and SHELL neurons. (**A**) Coefficient of variation (CV) of firing rate for CORE (gray) and SHELL (red) neurons during production of syllables that had high or low similarity to tutor syllables (top versus bottom 50% of tutor similarity scores). ***p<0.001 between CV during production of syllables representing high versus low similarity to tutor (Wilcoxon signed-rank tests; CORE n = 87, SHELL n = 105). (**B**) Characteristics of syllables in high- versus low-similarity categories. Left: fraction of syllable types represented in high versus low similarity to tutor syllables for each bird for day of singing; each symbol represents the number of syllable types produced in a recording session as a fraction of the total number of syllable types produced. Gray markers represent sessions in which activity of CORE neurons was collected (n = 17), red markers represent sessions in which activity of SHELL neurons was collected (n = 14), black markers represent sessions in which activity of both CORE and SHELL neurons was collected (n = 11). Right: the average scatter for each syllable type represented in high- versus low-tutor similarity categories for each session, defined as the average of the acoustic distance between each point in the distribution to its centroid (lower values indicate a tighter cluster). Box plots represent median and first/third quartiles.
DOI: https://doi.org/10.7554/eLife.26973.015

syllabic utterances. This result is reminiscent of the finding that juvenile birds exposed to adult females are able to produce more stereotyped song patterns than normally seen in young birds (*Kojima and Doupe, 2011*). The ability to produce syllable renditions that represent a range of matches to the goal (i.e. both higher and lower similarity) may be an important component of the behavioral variability that is necessary for skill acquisition involving reinforcement learning.

Only syllable utterances with low levels of tutor matching during subsong showed a progressive increase in tutor similarity over the course of development, and the activity of SHELL but not CORE neurons reflected this difference in behavioral trajectory. Developmental changes in firing rate variability mirrored that seen for behavioral development only in SHELL (*Figures 7* and *9*): firing rate CV in SHELL neurons decreased throughout development only among utterances that showed an increase in similarity over the course of learning. In contrast, variability of firing in CORE neurons did not change developmentally. The decrease in variable firing in SHELL neurons in parallel with the decrease in incidence of poorly matched syllable renditions indicates that the activity of SHELL neurons tracks the degree of song matching, consistent with the hypothesis that these neurons are involved in comparing song behavior to a tutor song memory.

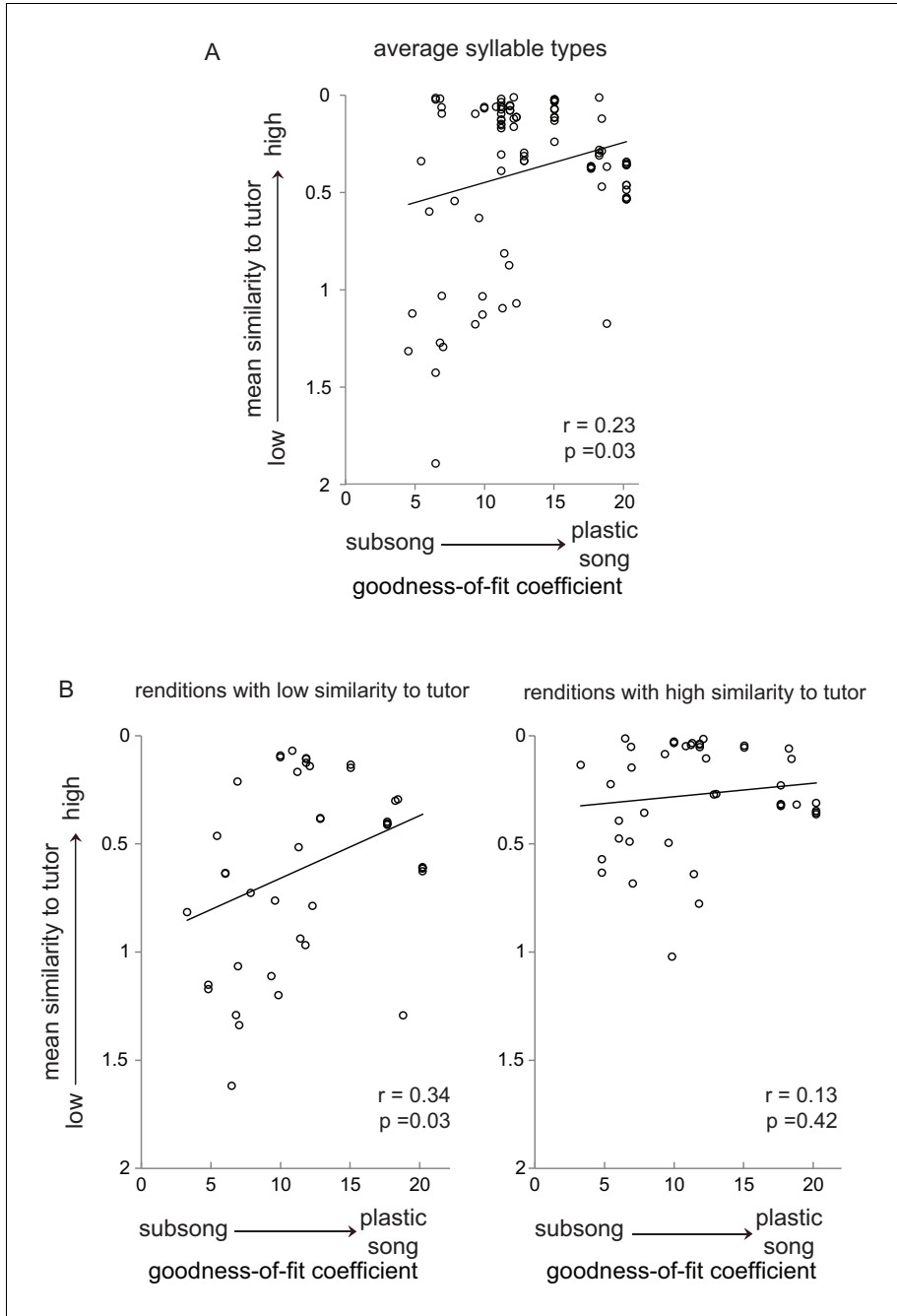

**Figure 7.** Subsets of syllable utterances during early sensorimotor learning had either high or low similarity to tutor syllables. (**A**) Average similarity of all juvenile syllable types to corresponding tutor syllables as a function of the progression of song development from subsong to plastic song (goodness-of-fit coefficients plotted against average syllable-type similarity for each bird for each day of singing; see *Figure 7—figure supplement 1*). (**B**) Average similarity to tutor across development, plotted as in A, but segregating individual syllable renditions into low similarity (bottom 50% of tutor similarity scores, left panel) and high similarity (top 50% of tutor similarity scores, right panel) for each day of singing for each bird. For tutor similarity scores, 2 = no similarity, 0 = perfect similarity (see Materials and methods).
DOI: https://doi.org/10.7554/eLife.26973.016

The following figure supplement is available for figure 7:

**Figure supplement 1.** Measuring degree of song development for each day of singing for each bird.
DOI: https://doi.org/10.7554/eLife.26973.017

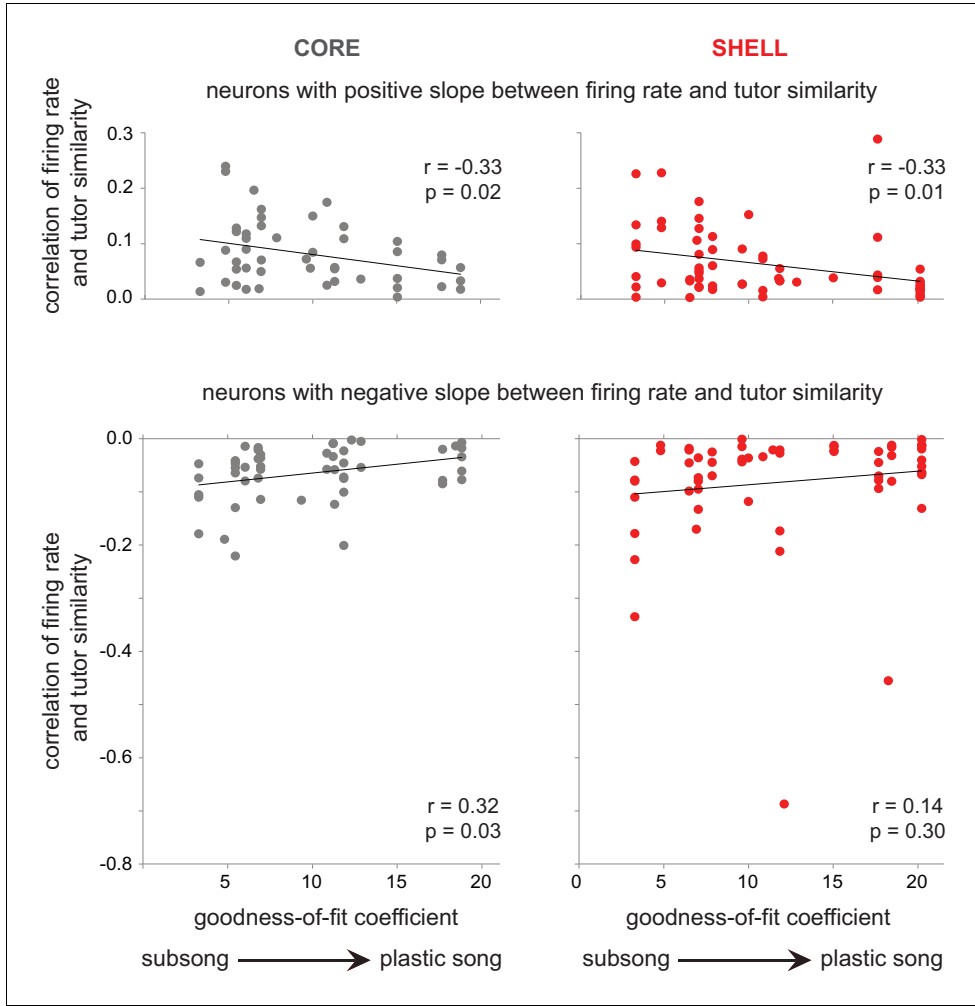

**Figure 8.** The association between firing rate and tutor similarity decreased in strength with the progression of song development. The correlation between baseline-corrected firing rate and tutor similarity (r values, y-axis) is plotted against degree of song development (goodness-of-fit coefficients, x-axis) for neurons with a positive slope between firing rate and tutor similarity (r values > 0; top panels) and for neurons with a negative slope between firing rate and tutor similarity (r values < 0; bottom panels). CORE (gray, left panels); SHELL (red, right panels).
DOI: https://doi.org/10.7554/eLife.26973.018

As vocal behavior was refined to achieve a more accurate imitation of tutor song, the correlation between tutor similarity and firing rate decreased in both CORE and SHELL neurons (*Figure 8*). The tendency toward weaker encoding of tutor similarity by firing rate as song learning progresses coincides with the loss of a large population of SHELL neurons that respond selectively to playback of tutor song (*Achiro and Bottjer, 2013*). Thus, tutor similarity is a better predictor of firing rate during early learning when behavioral variability is high and SHELL contains a large population of tutor-selective neurons. The loss of tutor-tuned neurons along with a progressive weakening of the association between firing rate and tutor similarity suggests that birds may be switching to a different strategy such as comparing their own utterances to a 'self' template more than to a tutor template. In accord with this idea, neural selectivity for self-generated sounds increases in CORE neurons between 45 and 60 dph (*Achiro and Bottjer, 2013*; *Doupe, 1997*; *Doupe and Solis, 1997*; *Solis and Doupe, 1997*). These patterns raise an important question: do individual SHELL neurons in which firing rate reflected tutor similarity (10.8%, *Table 2*) fall within the population of tutor-tuned neurons? Tutor-tuned neurons in juvenile LMAN-SHELL may act as a gate or filter for self-generated utterances that are tutor-similar, suggesting that evaluation of tutor similarity within LMAN may be selectively vested in the firing rate of neurons within this subtype during early sensorimotor integration.

Tutor similarity could be encoded by either increases or decreases in firing rate for both CORE and SHELL neurons. This intriguing aspect of the current results is subject to different interpretations. One possibility is that increases in firing rates for syllables with higher tutor similarity (positive slopes) could provide a reinforcement signal to increase the probability of producing that vocal pattern, whereas increases in firing rates for syllables with lower tutor similarity (negative slopes) could provide an error signal to decrease the probability of making that incorrect sound; both types of information could be conveyed to downstream neurons and used to guide accurate refinement of syllables during learning.

## Developmental aspects of skill learning

The construction of cortical circuits has been studied extensively in primary sensory cortex of mammals (*Katz and Shatz, 1996*; *Espinosa and Stryker, 2012*; *Levelt and Hübener, 2012*), but few studies have probed developmental changes in cortico-basal ganglia circuits as part of the mechanisms of skill learning. The involvement of LMAN circuitry in goal-oriented learning is likely to be strongly dependent on developmental changes that occur in these pathways. In juveniles, but not adults, many CORE neurons that project to vocal motor cortex send a collateral branch into the SHELL pathway (*Figure 1—figure supplement 3*) (*Miller-Sims and Bottjer, 2012*). This transient pathway may transmit a copy of the premotor signal generated by CORE neurons to SHELL neurons. In addition, the incidence of SHELL neurons tuned to learned tutor sounds decreases sharply (*Achiro and Bottjer, 2013*). Because the overall volume of SHELL regresses sharply during development, it may be that tutor-tuned neurons are eliminated due to naturally-occurring cell death (*Johnson and Bottjer, 1992*; *Johnson et al., 1995*), thereby helping to close the sensitive period for learning. Topographic specificity develops in the projection from LMAN CORE to RA (vocal motor cortex) during early development and is dependent on normal auditory experience (*Iyengar and Bottjer, 2002b*). In addition, a high proportion of synapses formed by thalamic axons in LMAN are 'silent synapses' in early development (*Bottjer, 2005*); a decrease in their number may contribute to a period of synaptic refinement to confer greater specificity of connectivity in thalamo-cortical connections (*Iyengar and Bottjer, 2002a*; *Nixdorf-Bergweiler, 2001*; cf. *Garst-Orozco et al., 2014*) as well as to curtail the sensitive period (*Huang et al., 2015*). These developmental changes suggest that differences in the extent and plasticity of neural mechanisms may be causally related to goal-oriented learning during development, and help to explain decreases in behavioral plasticity in older animals.

## Relationship of CORE and SHELL circuitry to mammalian cortico-basal ganglia pathways

A functional segregation of vocal learning into parallel CORE and SHELL cortico-basal ganglia loops is reminiscent of corresponding architecture in sensorimotor and associative cortico-basal ganglia loops of mammals. Studies in mammals have shown that neurons in associative loops show increased modulation early in learning of goal-directed tasks, whereas sensorimotor circuits increase their activity throughout training and may encode learned motor performance (*Joel and Weiner, 1997*; *Histed et al., 2009*; *Yin et al., 2009*; *Thorn et al., 2010*; *Gremel and Costa, 2013*; *Kim et al., 2013*; *Samejima and Doya, 2007*; *Graybiel, 2008*; *Yin et al., 2008*; *Ashby et al., 2010*; *Redgrave et al., 2010*; *Ito and Doya, 2015*; *Lehéricy et al., 2005*; *Thorn and Graybiel, 2014*; *Atallah et al., 2007*; *Nakahara et al., 2001*; *Parent and Hazrati, 1995*). Such evidence has suggested that associative loops function to evaluate motor performance during early stages of learning and sensorimotor loops encode behaviors as they become more habitual (*Makino et al., 2016*).

The overall patterns of neural activity we observed in SHELL and CORE are consistent with recent data showing that both associative and sensorimotor cortico-striatal circuits are engaged in skill acquisition, but that associative circuits disengage early in learning whereas sensorimotor circuits remain engaged (*Kupferschmidt et al., 2017*). The decline in spiking variability in SHELL neurons is consistent with a decreased involvement of associative circuits during early skill acquisition (*Thorn et al., 2010*; *Yin et al., 2009*). In addition, although variability of firing rate did not change developmentally in CORE neurons, firing rate CV was higher for syllable renditions with low tutor similarity in CORE neurons when averaged across all ages (*Figure 6*). This pattern suggests that CORE neurons may retain an exploratory (variable) mode (*Graybiel, 2005*), maintaining a higher CV for poorly-matched syllable renditions and a lower CV for well-matched renditions (*Figure 6*). Such an outcome

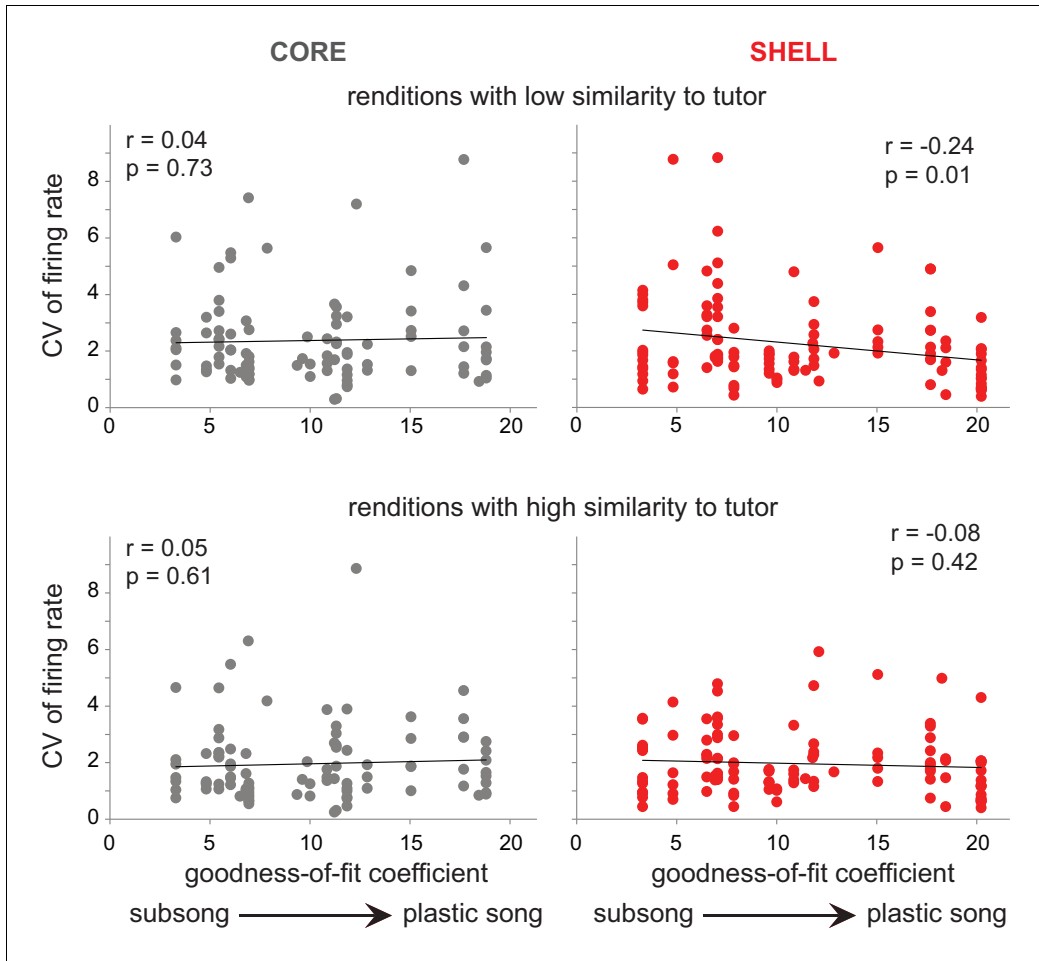

**Figure 9.** Variability of firing rate decreased in SHELL neurons during development for syllable renditions with low tutor similarity during subsong. Top panels: correlation of CV of firing rate during production of syllable renditions in the bottom 50% of tutor similarity with the progression of song development. Bottom panels: correlation of CV of firing rate during production of syllable renditions in the top 50% of tutor similarity with the progression of song development. Black lines indicate linear fits. CORE (gray, left panels); SHELL (red, right panels).
DOI: https://doi.org/10.7554/eLife.26973.019

would be consistent with the idea that CORE neurons retain some level of engagement at later stages of sensorimotor learning. Overall, the present results support the idea that motor learning across taxa entails the integrative product of multiple circuits whose functions reflect specific aspects during the progression of learning.

## Materials and methods

### Subjects

All procedures were performed in accordance with Protocol 9159 approved by the University of Southern California Animal Care and Use Committee and in accordance with the recommendations in the Guide for the Care and Use of Laboratory Animals of the National Institutes of Health. Ten juvenile male zebra finches (*Taeniopygia guttata*) were used, 43–60 days post hatch (dph) at the time of neural recordings (30 min recording sessions were collected across this age span on an average of 4 different days for each bird; thus each neuron represents a single 30 min recording session; see below). The period of sensorimotor integration begins in zebra finches when young males produce variable babbling sounds (subsong) starting at approximately 35 dph, and continues until the birds begin to produce a stereotyped imitation of the tutor song around 70–90 dph. Birds were

raised under naturalistic conditions (by their parents within group aviaries) until they were at least 38 dph, ensuring normal tutor song exposure and social experience (*Böhner, 1990*; *Immelmann, 1969*; *Roper and Zann, 2006*; *Böhner, 1983*; *Eales, 1985*; *Clayton, 1987*; *Mann and Slater, 1995*; *Chen et al., 2016*).

## Electrophysiology

Birds were anesthetized with isoflurane (1.5% inhalation) and an electrode assembly consisting of seven tungsten-wire stereotrodes was implanted into LMAN CORE and SHELL; each stereotrode consisted of two twisted-pair polyesteramide-insulated and imide-overcoated tungsten wires (diameter 25 μm, California Fine Wire Company, Grover Beach, CA) routed through fused silica capillary tubing. One stereotrode was implanted 1.5 mm dorsal to LMAN to serve as a reference electrode, and silver wire (diameter 250 μm) was placed between the skull and skin for the animal ground. Signals were acquired through a unity gain headstage (Neuralynx, Bozeman, MT) with a flexible cable that connected the electrode assembly to a commutator (Neuralynx) and two 8-channel amplifiers (Lynx8, Neuralynx). Vocalizations were recorded through a microphone mounted in the cage (Sanken COS-11D) and saved coincident with neural activity (band passed 300–6000 Hz; digitized at 32 kHz; Spike two software, Cambridge Electronic Design, UK). At the end of each experiment, small electrolytic lesions (7 μA for 20 s) were made to confirm recording locations. To verify the borders of LMAN CORE and SHELL, 50-μm-thick coronal sections were Nissl-stained or stained with a monoclonal antibody against calbindin-D-28K (Sigma-Aldrich Cat# C9848, RRID:AB_476894) using standard immunohistochemical procedures; calbindin expression specifically labels thalamic axons terminating in LMAN (*Pinaud et al., 2007*; *Achiro and Bottjer, 2013*) (*Figure 1*). The border between CORE and SHELL subregions of LMAN can be distinguished based on the density of magnocellular somata, and the outer borders of SHELL can be distinguished from surrounding regions based on the limit of calbindin expression (*Figure 1*). Recording sites were considered for analysis if they were confirmed histologically to be in either CORE or SHELL, excluding 50 μm on either side of the CORE-SHELL border (see *Achiro and Bottjer, 2013*).

Single units were isolated offline from stereotrode recordings made during each 30 min session using KlustaKwik (Ken Harris, Rutgers University) for automatic clustering as described previously (*Achiro and Bottjer, 2013*), and refined manually using MClust (A. David Redish, University of Minnesota) in MATLAB (Mathworks, Natick, MA, California). Single units were included if signal to noise ratio was $\geq 3$ and if less than 1% of spikes had an interspike interval <2 ms (n = 127 neurons from CORE, 171 neurons SHELL). To compute an appropriate sample size, we used preliminary electrophysiological data to measure average response strength that was different from zero during singing (including all neurons regardless of whether they showed a significant response during singing by t-test, see below). One-sample, two-tailed power analysis indicated we would need a sample size of 97 CORE neurons and 45 SHELL neurons in order to determine that neurons in each subregion showed a significant response during singing at 90% power with a 0.05 two-sided significance level:

$$n = \left( \sigma \frac{z\left(1 - \frac{\alpha}{2}\right) + z(1 - \beta)}{\mu - \mu_0} \right)^2, \ z = \frac{\mu - \mu_0}{\frac{\sigma}{\sqrt{n}}}$$

where n is sample size, μ is mean, $\mu_0$ is 0, σ is standard deviation, α is Type I error and 1 – β is power.

## Analysis of neural activity

Analyses of neural activity were based on all neurons from the total of all daily 30-min recording sessions across birds and days (each neuron was recorded for 30 min on 1 day). Baseline periods were defined as periods of silence (non-singing) lasting at least 2 s that were 2 s or more away from singing, calls or movement/cage noise. A neuron was considered responsive if the average firing rate (spikes/s) during singing showed a significant change from the average firing rate during baseline (independent t-test, due to differing number of baseline and singing episodes, p<0.05). Neurons in both CORE and SHELL showed significant modulation of firing rate during singing and/or the 50 ms interval prior to syllable onsets compared with quiet baseline periods (CORE: 81%, 103/127 neurons; SHELL: 82%, 141/171 neurons).

Analysis of pre-singing related activity was restricted to cells that showed a significant increase in average firing rate between baseline periods and the 50 ms prior to onsets of all syllables (independent t-tests, p<0.05). The proportions of CORE versus SHELL neurons showing an increase in average firing rate during this 50 ms period did not differ [CORE 48% (61/127 neurons), SHELL 40% (68/171 neurons); chi-square test = 2.27, p=0.13]. To analyze the temporal pattern of firing leading up to syllable onsets across these neurons, histograms of population activity were made by calculating the mean-subtracted firing rate for each neuron: the average firing rate was calculated during ±200 ms surrounding syllable onsets in bin sizes of 2 ms and the rate during each bin was subtracted from the mean firing rate over all bins for each cell before smoothing with a Gaussian (40 ms smoothing); population functions were generated by averaging across neurons (*Goldberg and Fee, 2012*). To determine significance, we used the inverse student t cumulative distribution function to get a t-statistic for 95% probability. We calculated the two-tailed critical value of the t distribution for α = 0.05 with degrees of freedom equal to the number of neurons minus one; a bin of the population histogram was deemed significant if the critical value of the t distribution multiplied by the population histogram ±the s.e.m. for that bin was less than or greater than zero.

To compare changes in activity during singing across neurons with differing firing rates, responses for each neuron were calculated as standardized response strength:

$$standardized\ response\ strength = \frac{\bar{S} - \bar{B}}{\sqrt{Var(S) + Var(B) - 2 * Covar(S,B)}} * \sqrt{n}$$

where S is the firing rate during singing, B is the firing rate during local baseline periods (the average of the two baseline periods nearest in time to each singing episode), and *n* is the number of singing/baseline pairs. A positive value indicates an increased firing rate during singing, and a negative value indicates a decreased firing rate during singing compared to baseline periods. We refer to this standardized measure throughout the text as response strength. To measure the incidence of bursts during singing episodes and local baselines, we calculated a burst fraction by measuring the percentage of spike events with an interspike interval of less than 10 ms (Wilcoxon signed-rank tests, due to one average local baseline period per singing episode,p<0.05).

## Analysis of singing behavior

Because juvenile birds in early stages of song learning do not produce stereotyped syllable sequences (song motifs), episodes of singing were defined as periods of continuous singing separated by gaps of at least 300 ms. Episodes of singing and individual syllables contained within episodes were detected automatically using amplitude threshold crossings and checked manually to remove cage noise and to adjust syllable start or stop boundaries where needed.

## Classification of syllable types

Juvenile birds produce highly variable sequences of syllables, making it impossible to align the temporal pattern of neural activity across song motifs (*Tchernichovski et al., 2001*). However, one can align spiking activity of single neurons to multiple renditions of individual syllables of the same type. Because of the high variability of juvenile syllables, we created custom software in MATLAB using many features created for Sound Analysis Pro (*Tchernichovski et al., 2000*) to automatically classify syllables in juvenile birds, available online (*Shen, 2017*; https://github.com/BottjerLab/Acoustic_Similarity. A copy is archived at https://github.com/elifesciences-publications/Acoustic_Similarity). To assign syllables to different types, we employed a combination of two measures of the acoustic distance between syllables that was then used to cluster syllables.

The first distance measure was based on summary statistics of the following syllable features: amplitude modulation, frequency modulation, center frequency, fundamental frequency, length, maximum frequency modulation within each frequency, maximum amplitude modulation across each frequency's power, pitch goodness (estimate of the amount of periodic energy), total power, derivative of total power, and Wiener entropy (estimate of spectral disorder). We calculated the following summary statistics for each feature (except length) over the duration of each syllable: mean, standard deviation, maximum, minimum, onset (average of samples 1–3%), middle (average of samples 49–51%), offset (average of samples 97–100%), correlation coefficient of the linear trend, time of maximum, and time of minimum. Each syllable was thus represented as a point in high-dimensional

space where each dimension was a summary statistic of one of the features (n = 101 feature values total). We then calculated the Euclidian distances between each point.

The second distance measure was based on time-varying changes in song features; we calculated the following syllable features for each time point in the syllable (9.27 ms window size, 7.91 ms overlap): Wiener entropy, frequency modulation, amplitude modulation, fundamental frequency and goodness of pitch (*Tchernichovski et al., 2000*)(*Figure 10A*). Syllables were represented as multidimensional feature vectors in time. The distance between the vectors was calculated with a dynamic time-warping algorithm including a warping penalty (*Vintsyuk, 1972*). The time-warping algorithm tolerates small perturbations in the timing of sub-syllabic events common to juvenile syllables.

In order to normalize the measures, we converted each distance measure to percentiles based on the empirical distribution (computed over all pairs of syllables). To obtain a combined distance measure we calculated the geometric mean of the two percentile distance measures for each syllable pair. We then generated a matrix consisting of the combined distance measure for each syllable to all other syllables. A final distance score was calculated as a dissimilarity index by taking one minus the correlation between points of the combined distance measure matrix, and ranged between 0 (perfect similarity) and 2 (no similarity; i.e., if the syllables were completely anticorrelated, the correlation would be −1 and the distance would be 1 - (−1)=2). Syllables were clustered into 4–25 types by hierarchical agglomerative clustering using the dissimilarity index as the distance metric and complete linkage as the linkage criterion (*Figure 10B*). We then manually selected the number of types for each bird for each day, confirmed cluster quality, and merged clusters which were similar. Some clusters were rejected due to high variability; thus not all syllables were clustered into types, especially those from recordings of birds producing more immature vocalizations. The total percent of classified syllables (assigned to clusters) was 49% overall (across song recordings for all birds and ages); 38% of syllables were classified from song recordings in early stages of development (representing the bottom half of song recordings by goodness-of-fit coefficients of exponential fit to syllable duration distributions, see 'Song development' below) and 57% of syllables were classified for song recordings in later stages of song development (representing the top half of song recordings by goodness-of-fit coefficient).

We calculated firing rates during each syllable produced and the 50 ms prior to syllable onsets in order to include pre-syllable-related activity; this syllable-based firing rate was used in analyses of neural selectivity for syllable types and similarity to tutor song (see below). The average gap between syllables in juvenile birds is ~60 ms (*Glaze and Troyer, 2013*; *Aronov et al., 2011*), making it unlikely that activity during a previous syllable was included in these firing rates. In order to align syllables to construct PSTH's of spiking activity, we linearly time warped each spike train to the average length for that syllable type following procedures of (*Kao et al., 2008*). We calculated the average duration of all syllables within each cluster and used that value as the 'reference duration'. Then, each syllable in the cluster was linearly stretched/compressed (syllables were aligned at onsets) and spike trains were projected onto the time-warped axis for each syllable. To determine selectivity of responses to specific syllable types, we calculated a sparseness/activity fraction (AF) (*Meliza and Margoliash, 2012*; *Vinje and Gallant, 2000*) for all neurons which responded significantly to at least one syllable type:

$$AF = \frac{1 - \left[ \left( \frac{\sum r_i}{n} \right)^2 / \sum \left( \frac{r_i^2}{n} \right) \right]}{1 - \frac{1}{n}}$$

where $r$ is the firing rate to the $i$th syllable type and $n$ is the number of syllable types. Thus a score of 0 indicates no selectivity for a specific syllable type (equal firing rates across all syllable types) and one indicates maximum selectivity (change in firing rate during only one syllable type).

## Similarity to tutor song

Analyses of similarity to tutor song included all neurons for which at least 40 classified syllables (syllables which were able to be clustered into a given type) were produced during each neuron's 30-min recording period. We used only classified syllable renditions for the tutor similarity analyses in order to use the same set of neurons and syllables for an analysis of prototypicality as a control (see below). We tested whether using only classified syllables influenced the tutor similarity results by

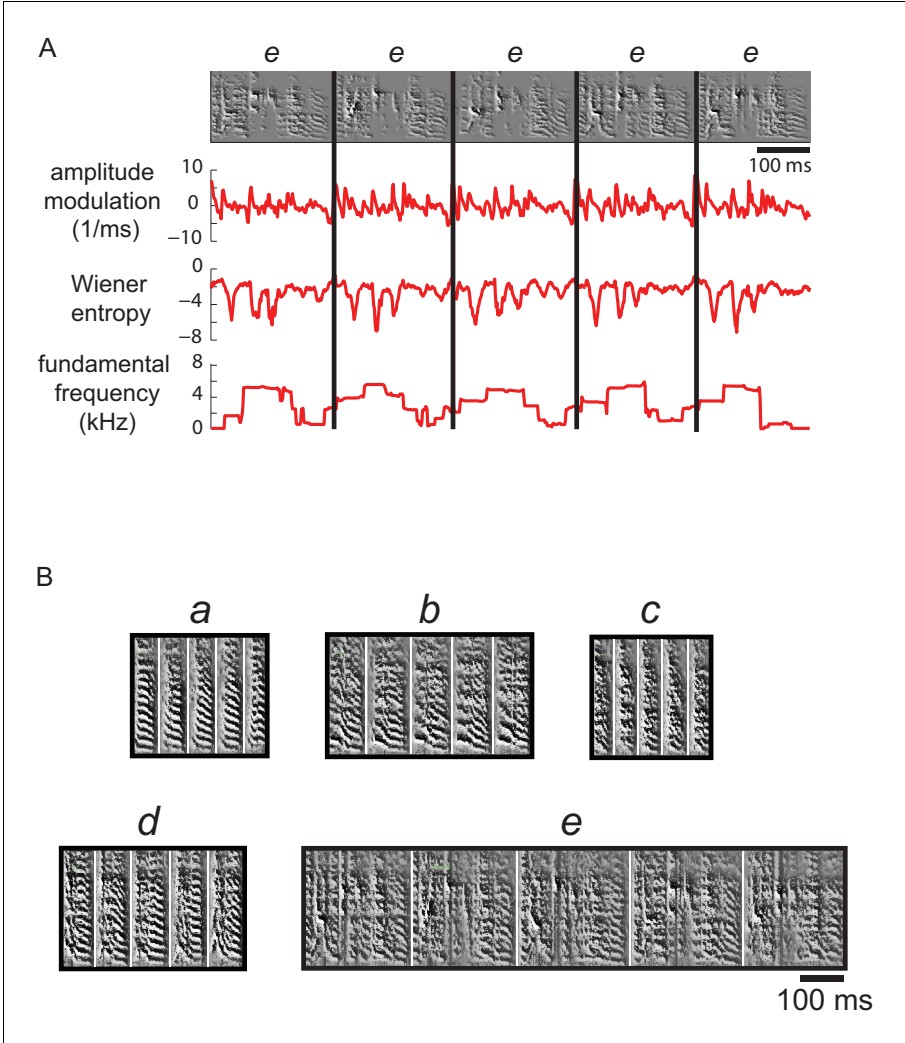

**Figure 10.** Multiple acoustic features were used to cluster syllable types. (**A**) Examples of the features used to calculate acoustic similarity in order to assign juvenile syllable renditions to types (clusters). Top row shows spectrograms for five renditions of a syllable type labeled e from a 59 dph bird (also shown in B). Below are plots of three features calculated across time for each syllable; a total of 11 features were used to generate an acoustic distance score for clustering of syllables into types (see Materials and methods). (**B**) Spectrograms for the five syllable types from this bird, a-e, resulting from automatic clustering of syllables.

DOI: https://doi.org/10.7554/eLife.26973.020

The following figure supplement is available for figure 10:

**Figure supplement 1.** Examples of syllable renditions with either high or low acoustic similarity to the closest-matching tutor syllable.

DOI: https://doi.org/10.7554/eLife.26973.021

examining the outcome using all syllables (both classified and not classified) and observed the same trends (data not shown); thus, use of classified syllables did not bias the results. These analyses included both singing-excited and singing-suppressed neurons, as well as neurons that did not show a significant change in firing rate during singing episodes (the latter cases ensured inclusion of cells that showed excitation during syllables with high similarity and suppression during syllables with low similarity, for example). To evaluate the similarity between juvenile syllables and tutor syllables we employed the final combined distance score described above. Similarity to tutor was calculated as the acoustic distance score between each syllable and its closest tutor syllable. We defined what constituted the closest tutor syllable in two ways: either as the tutor syllable which was closest in

distance to the center of the syllable cluster to which each rendition belonged, or as the tutor syllable which was simply closest in distance to each rendition. These two approaches yielded highly similar results, and we therefore chose the latter method to reduce any effects of the syllable clustering routine on tutor similarity calculations. All syllable renditions produced during each neuronal recording were ranked by tutor similarity; for many analyses, a median split was employed in which the ranked juvenile syllables for each neuron were divided into the top 50% versus bottom 50% of tutor similarity – these two categories are referred to as high and low similarity, respectively (*Figure 10— figure supplement 1*).

For each neuron, we calculated the linear regression between baseline-corrected firing rate during each syllable rendition and similarity to closest tutor syllable (*Figure 4B*). We used repeated permutation tests to estimate the fraction of neurons with significant correlations (*O'Connor et al., 2010*): for each neuron, we generated 1000 permutations by randomly shuffling the relationship between firing rate and tutor similarity and calculating the resultant r values. This provided a distribution of values under the null hypothesis of no relationship for each test. The fraction of neurons exceeding chance for each permutation test was computed as the fraction of actual r values falling above the. 975 percentile (positive correlations) or below the. 025 percentile (negative correlations) of the null distribution, and the average of this significant fraction across all permutation tests was taken as the fraction of neurons with significant r values.

To assess the association of baseline-corrected firing rates on tutor similarity at the population level, we employed a mixed-effects linear regression model of baseline-corrected firing rates across all neurons with fixed and random effects for tutor similarity nested within a random intercept for neurons using an unstructured covariance matrix; because firing rate was measured at multiple tutor similarities within each neuron, our model fit a random intercept for neurons, with tutor similarity treated as both a fixed and random slope.

To examine if firing rate variability was modulated by similarity to tutor song, we calculated the CV of firing rate during production of syllables representing high and low similarity to corresponding tutor syllables (top and bottom 50% of syllables ranked by tutor similarity). Neurons were included in these analyses if at least 40 classified syllables were produced during the 30-min recording period and if the mean firing rate was non-zero for the most/least prototypical and most/least similar to tutor song syllables (because neurons with mean firing rates of zero would give undefined CV values).

## Syllable prototypicality

We calculated a prototypicality score for each syllable based on (*Niziolek et al., 2013*), which measures whether renditions are similar to the center of that syllable's distribution (more prototypical) or less similar to the center of the distribution (less prototypical). We computed the acoustic distance between each syllable rendition and the center of the syllable cluster to which it belonged, again employing the combined distance score described above. This measure served as a control for effects based on tutor similarity to assess whether correlations based on firing rate reflected prototypical utterances; we used the same mixed-effects regression model as above to assess a relationship between baseline-corrected firing rates and prototypicality. As indicated above, this analysis also included all neurons for which at least 40 classified syllables were produced during individual recording periods; in this way analyses of tutor similarity and prototypicality included the same set of neurons and syllables.

## Song development

In order to assess how developed each bird's song was for each day, we utilized methods previously described to define song stage (*Aronov et al., 2011*). Juvenile birds in the subsong stage of sensorimotor integration produce syllables of variable lengths, the distribution of which is well-fitted by an exponential function. As birds progress to the plastic song phase, they produce more regular syllable types which begin to appear as peaks in the distribution of syllable durations, and therefore are no longer well-fit by an exponential (*Aronov et al., 2011*; *Tchernichovski et al., 2004*). Based on this evidence, we fit an exponential to all syllable duration distributions for each bird, for each day of singing (500–8,000 syllables). We used the Lilliefors test (MATLAB) to quantify goodness-of-fit, and the resulting coefficient was scaled by the number of syllables. Therefore, smaller goodness-of-fit

coefficients indicate less developed songs (i.e. subsong) and larger coefficients indicate more developed songs (i.e. plastic song). Response strength, CV, and r values (correlations between baseline-corrected firing rate and tutor similarity) for all neurons across all classified syllables (syllables assigned to clusters) produced in each recording session were analyzed as a function of song development using goodness-of-fit coefficients.

## Statistics

Kolmogorov-Smirnov and Shapiro-Wilk tests were used to test for normality; t-tests were used to compare means for normally distributed data, and Mann-Whitney tests were used for non-normally distributed data. Differences in proportions were tested using chi-square tests. Correlations were performed using Pearson's correlation. The significance of mean-subtracted bins for pre-singing activity (*Figure 3*) was calculated as the 95% confidence interval outside of zero and is described above. Specific statistical tests used are identified in context in the Results.

## Acknowledgements

This work was supported by NINDS grants NS 037547 and NS 087506, NIDCD Training Grant DC 009975, and NINDS Training Fellowship NS 073323. The authors declare no competing financial interests. We thank Rachel Yuan for comments on the manuscript, Arthur Shau for technical assistance, and Nicholas Jackson for expert statistical advice.

## Additional information

### Funding

| Funder | Grant reference number | Author |
|---|---|---|
| NIH Office of the Director | Research grant NS087506 | Sarah Bottjer |
| NIH Office of the Director | Training grant DC009975 | Sarah Bottjer |
| NIH Office of the Director | Training fellowship NS 073323 | Jennifer M Achiro |
| NIH Office of the Director | Research grant 037547 | Sarah Bottjer |

The funders had no role in study design, data collection and interpretation, or the decision to submit the work for publication.

### Author contributions

Jennifer M Achiro, Conceptualization, Data curation, Formal analysis, Validation, Visualization, Writing—original draft; John Shen, Software; Sarah W Bottjer, Conceptualization, Supervision, Funding acquisition, Project administration, Writing—review and editing

### Author ORCIDs

Jennifer M Achiro (iD) http://orcid.org/0000-0002-3978-1647
Sarah W Bottjer (iD) http://orcid.org/0000-0003-4365-9166

### Ethics

Animal experimentation: All procedures were performed in accordance with Protocol #9159 approved by the University of Southern California Animal Care and Use Committee and in accordance with the recommendations in the Guide for the Care and Use of Laboratory Animals of the National Institutes of Health.

### Decision letter and Author response

Decision letter https://doi.org/10.7554/eLife.26973.024
Author response https://doi.org/10.7554/eLife.26973.025

## Additional files

**Supplementary files**
• Transparent reporting form
DOI: https://doi.org/10.7554/eLife.26973.022

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
