## [Decision Letter]

[Editors’ note: this article was originally rejected after discussions between the reviewers, but the authors were invited to resubmit after an appeal against the decision.]

Thank you for submitting your work entitled "Neural activity in corticobasal ganglia circuits of juvenile songbirds encodes performance during goal-directed learning" for consideration by *eLife*. Your article has been reviewed by three peer reviewers, one of whom, Ronald L Calabrese (Reviewer #1), is a member of our Board of Reviewing Editors, and the evaluation has been overseen by a Senior Editor..

Our decision has been reached after consultation between the reviewers. Based on these discussions and the individual reviews below, we regret to inform you that your work will not be considered further for publication in *eLife*.

This is an interesting manuscript, which compares the properties and developmental changes of neuronal activity in cortico-basal ganglia circuits of the zebra finch during the juvenile song learning period. This work focuses on the well-studied cortical LMAN CORE and on the surrounding SHELL. The hypothesis that individual LMAN neurons may function to compare sensory feedback and efference copy from current motor actions to the song target, with SHELL serving as the critic and CORE as the actor, is very appealing.

Major Concerns

1) This is mainly a correlative study here without any attempt to dissociate sensory from motor effects on firing.

2) There were substantial concerns about the appropriateness and correctness of the statistical analyses. The correlative nature of the study makes appropriate statistics critical.

3) The analyses of the data are very complicated and somewhat idiosyncratic to songbirds and their unique song-learning behavior. This make for a difficult read for a non-expert.

4) The issue of overwhelming details and smallness of effect concerned us. The idea of the paper is appealing, but we were not overly convinced by their argument. This seems particularly important while considering a manuscript in a journal with a general readership.

*Reviewer #1:*

This is an interesting manuscript, which compares the properties and developmental changes of neuronal activity in cortico-basal ganglia circuits of the zebra finch during the juvenile song learning period. During this period, juvenile songbirds are actively engaged in evaluating feedback of self-generated behavior in relation to their memorized tutor song (the goal). This work focuses on the well-studied cortical LMAN CORE and on the surrounding SHELL, which the corresponding author's group has delineated in several previous studies and implicated in the process of learning during song development of juveniles. It also tracks development of songs themselves during this period of song learning so that neuronal responses and songs can be compared as learning progresses. The paper uses rather complicated and multifaceted analyses of songs and of responses of CORE and SHELL neurons recorded in awake singing juvenile birds, which nevertheless present a consistent picture. The firing rate of a subset neurons in both SHELL and CORE reflect the degree of tutor song matching during singing, with a higher incidence of SHELL neurons showing correlations between firing rate and degree of tutor song matching compared to those in CORE. In addition, variability in firing rate was lower during production of the best-matched syllable utterances compared to poorly-matched utterances. Syllable utterances least similar to tutor syllables showed a progressive increase in similarity to the tutor over the course of vocal learning and the activity of SHELL, but not CORE neurons reflected this difference in learning trajectory between best and worst tutor-matched syllable renditions. These results suggest that individual LMAN neurons may function to compare sensory feedback and efference copy from current motor actions to the song target, with Shell serving as the critic and CORE as the actor.

This paper was difficult to read because the analyses are complicated and somewhat abstract, but the text is tightly written.

Major Concerns

1) The analyses of the data is very complicated and somewhat idiosyncratic to songbirds and their unique song-learning behavior. This make for a difficult read for a non-expert.

2) This reviewer cannot fully critique the data analytical methods but defers to the other experts. One point that make this reviewer particularly uncomfortable is the use of correlations, as in Figure 13 (but others also), which are critical for the central conclusion concerning developmental progression in the responses of SHELL neurons. The data has virtually no vertical spread and thus r will be very small and difficult to obtain significance or show substantive change. Moreover, I am uncomfortable in general that these analyses focus on the outer quartiles of the distributions only (most similar and least similar groups).

*Reviewer #2:*

This paper explores important issues regarding sensorimotor processing during vocal learning. However serious problems in the quantitative analyses prevent a clear interpretation of the data.

First, many analyses rest on comparing the proportions of statistical tests that achieve significance at p<0.05. Although such approaches can be useful, in many cases the authors do not perform the correct control analyses to support their claims. For example, the authors cite the fact that between 6-10% of correlations are significant as evidence that "neural activity is correlated with similarity to tutor song in both CORE and SHELL subregions of juvenile LMAN". This conclusion is not supported by the data unless the authors show that, for example, finding that 7% of CORE cells have a different firing rate for the most/least similar syllables is significantly different from chance. That is, running 1,000 t-tests on vectors of random numbers will, on average, produce significant correlations (at p<0.05) in 5% of cases. To provide evidence for their conclusions here the authors must demonstrate that finding 7% of correlations significant is surprising.

Additionally, the analysis presented in the section "Activity correlated with multiple, but not single, features" is not convincing. It is not clear to me whether the basic finding – that many neurons have a significant relationship with 11 features simultaneously, but few have sig relationships in single regression – is actually informative. First, finding a significant correlation with the 11 "aggregate features" might simply mean that activity is correlated with one, rather than several, of the features. It would therefore be unsurprising if a far smaller number of single-feature tests were significant. More importantly, and completely separate from the above concern, there is no guarantee that the acoustic features measured are the features that the bird actually evaluates or controls, so the conclusion about representing individual vs constellations of features is unconvincing, i.e. even if the individual vs aggregate analysis were performed correctly, it could be that the LMAN neurons really do care about single features, but that these features are different from, although possibly correlated with, the 11 features used in the analysis.

The logic of the analysis used in Figure 4 is confusing. As I understand it, the authors first select neurons that fire greater than baseline rate in the 50 msec prior to syllable onset, and THEN ask whether there is a change in average activity change locked to syl onset. (Such data are used to argue that only CORE cells shows "coordinated premotor activity"). But isn't it the case that to be included in this analysis, SHELL neurons by definition must have elevated rates (relative to baseline) that stay elevated throughout syllable onset, rather than dropping/modulating as is apparently typical in CORE cells? If so, it does not seem correct to say that SHELL neurons do not display coordinated premotor activity, or do so to a lesser extend than do CORE cells.

A further fundamental problem is that no attempt is made to separate the sensory from motor tunings of these neurons. All other problems with the analysis aside, finding that neural activity co-varies with syllable acoustics immediately raises the question of whether this covariation reflects premotor control (or efference copy) of future behavior, sensory processing of prior behavior, or some combination thereof. This issue confounds interpretation of most of the results in this paper. For example, the finding that variability in neural "responses" is correlated with similarity to tutor (subsection “Variability in neural responses of both CORE and SHELL neurons correlates with similarity to tutor song but not with syllable prototypicality”) might reflect either auditory or motor differences between the syllables that are most- or least-similar to those of the tutor. Given this concern, and the numerous problems with the quantitative analyses outlined above, it is not clear to me that this paper significantly advances our understanding of sensorimotor processing.

*Reviewer #3:*

The paper by Achiro, Chen and Bottjer shows evidence for neurons in a motor learning pathway that seem to encode the motor performance during learning. This finding is important as it provides evidence for a goal-directed signal that has been an elusive, to date. This has important implications for investigators studying motor learning, particularly those interested in the role of the basal ganglia in goal-directed behaviors. Overall, this manuscript is a tour de force that combines difficult (hard to get) neural recordings with a great deal of careful analysis. As far as I get tell the statistical analyses are good. Although the analysis is very thorough, the paper seems to suffer under the weight of complexities of the analysis so that as I through the paper the analysis became so detailed that I lost the narrative of the paper. This could be particularly problematic for non-birdsong readers. I am not sure how to fix the problem as most of the analysis seems important to 'make the case' and the authors summarize each section.

1) The results are correlative and there is no manipulation of the system to confirm that results. This requires additional experiments that are far beyond the scope of the current results, although they would add additional support to the important finding.

2) In addition to the analysis, the effect seems to be quite small; only a small percentage of neurons and a small-ish change in firing rate (Figure 8). I do not have a problem with the small percentage of neurons as the bird may not need many neurons to encode the performance, particularly if those neurons 'disappear' after learning. This does not mean it is not true, or unbelievable, it just makes it harder to understand and show easily. For example, the firing rate increases are small and the similarity modulation index shift is very hard to see from the graphs, although there is a statistical shift.

3) I think it would be useful to show some raw (at least spikes) data for Figure 2. In the figure it is difficult to tell if there is a change in firing rate from baseline as not much baseline is shown and for some of the songs, the firing rate does not seem to change, for example, in the first LMAN shell singing-excited neuron shown, the neuron seems to fire at the same rate before and during song, except for a burst of activity at song onset. In other words, if I looked at the firing rate I would not be able to tell when the bird song. One way this could be addressed is by providing more baseline activity for one song in each category.

4) In subsection “Neural responses in juvenile LMAN are variable during repeated renditions of emerging syllable types “, the authors say they time warped the spike train to look at timing. This seems to be a little tricky to warp the timing to look at timing. How much did the timing have to be warped (% of syllable duration)? How much warping occurred compared to the variability of the spiking activity?

5) I found the Similarity Modulation Index graphs a little confusing (Figure 8 and Figure 9). Where the neurons with positive response strengths and negative response strengths analyzed together? If you separate them, do the positive responses result in a significant result and negative responses result in a non-significant results?

[Editors’ note: what now follows is the decision letter after the authors submitted a revised manuscript for consideration.]

Thank you for submitting your article "Neural activity in corticobasal ganglia circuits of juvenile songbirds encodes performance during goal-directed learning" for consideration by *eLife*. Your article has been reviewed by three peer reviewers, one of whom, Ronald L Calabrese (Reviewer #1), is a member of our Board of Reviewing Editors, and the evaluation has been overseen Andrew King as the Senior Editor. We had two new referees.

The reviewers have discussed the reviews with one another and the Reviewing Editor has drafted this decision to help you prepare a revised submission. Given that this paper has gone through two full rounds of review and remains in an unacceptable form, we are prepared to offer only one more opportunity to provide an acceptable version of the manuscript.

Summary:

This interesting manuscript compares the properties and developmental changes of neuronal activity in cortico-basal ganglia circuits of the zebra finch during the juvenile song leaning period. During this period, juvenile songbirds are actively engaged in evaluating feedback of self-generated behavior in relation to their memorized tutor song (the goal). This work focuses on the well-studied cortical LMAN CORE and on the surrounding SHELL, which the corresponding author's group has delineated in several previous studied and implicated in the process of learning during song development of juveniles. It also tracks development of songs themselves during this period of song learning so that neuronal responses and songs can be compared as learning progresses. The paper uses sophisticated analyses of songs and of responses of CORE and SHELL neurons recorded in awake singing juvenile birds, which present a consistent picture. The spiking patterns of a subset neurons in both SHELL and CORE reflect the degree of tutor song matching during singing. Both CORE and SHELL neurons encode tutor similarity either by increases or decreases in firing rate, but only SHELL neurons showed a significant association at the population level. During development, tutor similarity (for syllables with low initial similarity to tutor syllables) predicted firing rates most strongly during early stages of learning, and SHELL but not CORE neurons showed decreases in response variability, suggesting that the activity of SHELL neurons reflects the progression of learning. These results suggest that individual LMAN neurons may function to compare sensory feedback and efference copy from current motor actions, with Shell serving as the critic and CORE as the actor.

Essential revisions:

This paper was difficult to read because the analyses are complicated and somewhat abstract, and many separate issues are addressed which are peripheral to the main interest of the paper; during development, tutor similarity predicted firing rates most strongly during early stages of learning, and SHELL but not CORE neurons showed decreases in response variability. This paper has thus much inherent interest but is too unwieldly as presented and how it should be revised was the subject of lively constructive discussion by the expert reviewers. In revision, the authors should focus and provide detailed responses to the points below.

1) The main revision must be to simplify the message of the paper by focusing ONLY on the song similarity analysis.a) The 'premotor vs non premotor' analyses are not necessary to support the main finding, and given the relatively weak support in favor of a CORE-SHELL distinction on that issue, it should be removed.b) A large portion of the results is devoted to showing that juvenile syllables become more and more similar to tutor during early song development. Although previous studies have focused on the later stages of learning, song similarity is known to increase even in the early stages of learning (e.g. Aronov et al., 2008). The authors should cut substantially the lengthy description and analysis of song development to concentrate on the core issues; during development, tutor similarity predicted firing rates most strongly during early stages of learning, and SHELL but not CORE neurons showed decreases in response variability.c) The concentration on the relation between LMAN firing and the acoustic properties of the syllables is unwarranted. The issue here is that findings are very weak. The authors make a very problematic statistical claim about representation of multiple features but no single feature. This could be, in principle, true, but the analyses show simply a very weak effect, as opposed to synergy or gestalt representation. It is not possible to see anything interesting in the raw data. To quote one of the expert reviewers in discussion "The entire section about analysis of acoustic features, and all figures showing nothing apparent, should be removed, and replaced by something like "we found a very weak, but statistically significant representation of song features (t=… p=… see suppl.)."d) Another issue that undermines the clarity of the paper is the intermixing of results that apply both to LMAN-core and LMAN-shell neurons and results that contrast between these two areas. The authors should highlight similarities and differences between these two areas. As suggested by the first sentence in the discussion, the main finding of the paper applies to both core and shell neurons, and only minor differences are found between the singing-related firing in these two populations. This could be made more explicit throughout the results.

[Editors' note: further revisions were requested prior to acceptance, as described below.]

Thank you for resubmitting your work entitled "Neural activity in corticobasal ganglia circuits of juvenile songbirds encodes performance during goal-directed learning" for further consideration at *eLife*. Your revised article has been favorably evaluated by Andrew King (Senior editor), a Reviewing editor, and two reviewers.

The manuscript has been improved but there are some remaining issues that need to be addressed before acceptance, as outlined below:

The manuscript is much improved and all comments were fully addressed.

There are two remaining issues that should be resolved before this paper is published.

1) Regressions of baseline-corrected firing rates against tutor similarity for each neuron revealed that cells in both CORE and SHELL exhibited either positive or negative associations between neural activity and degree of tutor similarity: approximately half of all neurons in each subregion showed positive slopes (r values > 0, increased firing rates for syllables with higher tutor similarity).

We should have noticed this issue in the previous round, but this description is deficient. Saying that half increased and half decreased is like looking at random data, right? Instead, plot a histogram of all those slopes and/or r values, so readers can see the distribution on both positive and negative sides. Then do bootstrap (shuffling tutor similarity) to plot this histogram against a random distribution of similarities. This way, it would be possible to evaluate if this is a real correlation.

2) Response strength did not differ between CORE and SHELL neurons for either excitation or suppression (Table 1; Mann-Whitney tests: singing-excited neurons U = 1768, p = 0.06, singing-suppressed neurons U = 459, p = 0.90).

Maybe you should not say that there is no difference for singing excited neurons since 0.06 is almost significant. Instead, say that you see a borderline difference between core and shell, which did not reach significance level, for song-excited neurons, but no apparent effect in the song-suppressed neurons and then give the p values.

If the statistical tests in comment #1 work out satisfactorily, then further review by the expert reviewers will not be required.

---

## [Author Response]

[Editors’ note: the author responses to the first round of peer review follow.]

This is an interesting manuscript, which compares the properties and developmental changes of neuronal activity in cortico-basal ganglia circuits of the zebra finch during the juvenile song learning period. This work focuses on the well-studied cortical LMAN CORE and on the surrounding SHELL. The hypothesis that individual LMAN neurons may function to compare sensory feedback and efference copy from current motor actions to the song target, with SHELL serving as the critic and CORE as the actor, is very appealing.

All three reviewers expressed some strongly positive comments about the paper. All reviewers also made important comments, many of which we agree with and have used to greatly improve the analysis of the Results and the exposition of the paper overall. We respectfully disagree with some of the concerns expressed by the reviewers.

Major Concerns1) This is mainly a correlative study here without any attempt to dissociate sensory from motor effects on firing.2) There were substantial concerns about the appropriateness and correctness of the statistical analyses. The correlative nature of the study makes appropriate statistics critical.3) The analyses of the data are very complicated and somewhat idiosyncratic to songbirds and their unique song-learning behavior. This make for a difficult read for a non-expert.4) The issue of overwhelming details and smallness of effect concerned us. The idea of the paper is appealing, but we were not overly convinced by their argument. This seems particularly important while considering a manuscript in a journal with a general readership.

One major concern is that “this is mainly a correlative study”. This is true, but we disagree strongly with the premise that correlational studies cannot provide ground-breaking data, as this paper does. Many recent correlative studies have used the approach of chronic recordings in awake behaving animals, as we did here, to relate spiking activity to behavior and provide major new data. This is especially true for studies that have recorded the activity of cortical neurons during expression of a specific behavior; given the diversity of cortical neurons, this approach constitutes a major first step in understanding how cortical neurons mediate complex behaviors. One recent example (among many) comes from barrel cortex: Hires, Gutnisky, Yu, O’Connor, & Svoboda (2015, *eLife*, 4:e06619). This paper recorded activity in barrel cortex during an object localization task. In that study, as in ours, variations in a naturally-occurring behavior provide the contrast that is needed to make important inferences about neural function. In our study, variations in the similarity of self-generated behavior to memorized tutor sounds across the sensorimotor stage of learning was used as a predictor of neural activity. The results are the first to identify learning-related signals as juvenile songbirds are refining their behavioral output.

Specifically, our study provides the first direct evidence that the cortical region LMAN contains neurons that are involved in coding similarity between self-generated and goal behaviors. I published a paper in 1984 showing (based on lesion data) that LMAN plays an essential role in juvenile birds as they are actively engaged in learning. Since then, many songbird labs have looked for evidence that LMAN neurons in juveniles mediate song learning based on comparing vocal-auditory feedback to an internal representation of tutor song. The few studies that have tested this idea directly have provided negative evidence; for example, studies that attempted to disrupt auditory feedback during singing observed no difference in the spiking patterns of LMAN neurons (e.g., Leonardo, 2004, PNAS, 101:16935). The goal of our study was to test the hypothesis that LMAN neurons would encode variations in the similarity of on-going vocal production to corresponding tutor syllables. No previous study has attempted to quantify similarity between juvenile syllables and learned tutor syllables during early stages of sensorimotor integration; we accomplished this by further developing a program that has been universally used among songbird labs to analyze acoustic features (Sound Analysis Pro, Tchernichovski et al., 2000). In addition we carried out the difficult technical challenge of recording from vocalizing birds at this young age. We found strong evidence that the activity of LMAN neurons does encode similarity of self-generated behavior to tutor behavior, indicating a direct participation in mediating goal-directed learning, especially in LMAN-SHELL. This is the first and only evidence to this effect in the past thirty years, and constitutes an important break-through. In the songbird system in particular, where very little is known concerning the organization and physiology of cortical neurons, studies that measure spiking activity in relation to on-going behavior, represent necessary and important steps to advancing mechanistic understanding and formulating specific hypotheses for further tests.

A corollary point is that we disagree strongly that separating sensory from motor tuning, or any “manipulation of the system” is a necessary part of this study. In our opinion, the current data are a goldmine which begin to inform mechanisms and serve as an essential prerequisite for many future tests, including questions regarding whether the tutor-matching activity represents efference copy, auditory feedback, both, or other influences. Thus, the learning-related signals we report are likely to reflect a complex combination of sensory and motor signals; in the recurrent loop architecture of cortico-basal ganglia circuits these signals will be iteratively conveyed both through and across the types of parallel pathways we describe. We favor a combination of influences compatible with both forward and inverse models; that is, this neural circuitry is likely to entail efference copy learning to predict sensory feedback as well as sensory inputs learning to predict the motor commands that gave rise to them. The data in this paper will be extremely useful for developing new computational models of learning in this system to guide the many future experiments required to dissect these important questions.

The fact that the results provided strong evidence in favor of “outcome evaluation” between self-generated and tutor behavior is surprising, given that we know so little about the microcircuitry of LMAN and the heterogeneity of neurons contained therein – i.e., we were looking for the proverbial needle in a haystack. The single neurons that showed a significant association between tutor syllables and self-generated syllables might easily fall within a subtype of SHELL neurons, in which case the percentage of significant neurons would be much higher. As we comment in the Discussion, we think it is likely that single neurons that signal tutor similarity may be vested within the population of SHELL neurons that respond only to playback of tutor song (Achiro and Bottjer, 2013). But in any case, as pointed out by Reviewer #3, a small number of neurons can go a long way. The famous cholinergic interneurons of the striatum constitute only 0.3% of the total neuron number, but are important for procedural learning (Tepper & Bolam, 2004). This may be difficult to understand but does not detract from the importance of the effect.

Regarding the “smalli-ish change in firing rate”: LMAN neurons, like many cortical neurons, are sparsely firing. This has been observed in all published electrophysiological studies of LMAN (e.g., see Achiro and Bottjer, 2013, and references therein). The range of firing rates in any population of sparsely firing neurons will be limited. One could argue that it might be less likely to observe significant correlations for this reason, but we observed highly significant correlations between firing rate and tutor similarity. We have clarified this in the text.

We have stream-lined the analyses of the data and the presentation of the Results considerably so that the paper is much easier to read. Based on the concerns of reviewers concerning the statistical analyses, we sought expert statistical advice. Reviewer #1’s concern that we based many analyses on the top and bottom quartiles of tutor similarity turned out to be justified. We have eliminated this approach altogether, and for the main result of the paper (prediction of firing rate by tutor similarity), we substituted a mixed-effects regression analysis that employed all the data. This analysis yielded a main effect of tutor similarity in SHELL but not CORE neurons. This analysis also caused us to focus on the fact that single neurons could show either positive or negative associations between firing rate and similarity of self-generated utterance to tutor syllables; assessing the magnitude of positive versus negative correlations within each subregion (as descriptive data, since this was not a prediction of our study) clearly indicated that firing rate in CORE neurons is modulated by tutor similarity, albeit to a lesser degree than that of SHELL neurons. Thus the primary result of the paper did not change, although now the data include the added dimension that tutor similarity can be encoded by either increases or decreases in firing rate.

Reviewer #2 made an excellent point regarding the fact that many analyses rested on proportions of statistical tests of single neurons that achieve significance at p<.05. We agree and have altered that approach in two important ways. First, by applying correction factors for judging the proportion of single neurons (again, based on expert statistical advice). Secondly, for testing the percent of single neurons that encode tutor similarity, we performed permutation tests in which 1,000 random shuffles (of firing rate relative to tutor similarity) were performed for each neuron; r values were calculated for each random shuffle to provide a distribution of null values. For each permutation, the fraction of actual neurons exceeding chance (0.025/0.975) was calculated, and the average of this significant fraction across all permutations was taken as the fraction of neurons with significant r values. This approach fully addresses the concern of Reviewer #2 regarding whether 10% of neurons is “surprising” (i.e., significant), and produced only minor changes in the percentage of single neurons: 5.5% of CORE neurons and 10.8% of SHELL neurons exhibited a significant correlation of firing rate to degree of tutor song matching (previously these values were 6% and 10%). However, when we performed this same analysis on the prototypicality scores, the fraction of significant neurons dropped in SHELL neurons but increased slightly in CORE neurons, which eliminated the significant difference between them. Given this (and the fact that prototypicality showed no significant effect at the population level), we now employ prototypicality purely as a control for tutor similarity. This approach simplifies the exposition and makes the paper much easier to absorb for general readers.

We disagree with Reviewer #2 that the section entitled “Neural activity is correlated with multiple, but not single, acoustic features of syllables” is not convincing. This reviewer suggests that the correlation of a neuron with multiple acoustic features could be based on a strong correlation with a single feature; if this were true, then that single feature would have shown a strong correlation, but that was not the case. That is, single-feature tests could not explain the correlation seen in single neurons. The GLM (a General Linear Model – not a Generalized Linear Model) used in this analysis is simply a linear regression including all acoustic features, and includes partial regressions for each family of features (i.e., all summary statistics for each acoustic feature); no correction for multiple comparisons is applied to the partial regressions for single features since the analysis for single features already takes the other features into account. Our expert statistician confirmed that this analysis is fully appropriate. He also suggested that we replace the Bonferroni correction that we had applied to judge the percent of significant cells with an FDR (False Discovery Rate) correction in this case, which we have done.

This reviewer also commented that we may not have measured acoustic features that the bird actually evaluates or controls. This is completely true, and we have added this caveat to the manuscript. However, the acoustic features we measured have been in use in most songbird labs since 2000, when they were originally developed by Ofer Tchernichovski and Partha Mitra; it would be difficult to publish any songbird paper that quantified acoustic features without the use of such analyses. We have used 100% of the features used in their software (SAP, Sound Analysis Pro), and have added other features and measures. Of course this does not mean we have included the “right” features, but we have carried out one of the most complete acoustic analyses that we know of to date. Furthermore, we found that the acoustic features we measured predict the firing rate in a large percentage of LMAN neurons. It was this approach that enabled us to successfully analyze syllables produced by birds as young as 43-50 days, which no previous paper has done.

Lastly, we agree that the paper was very difficult to read and absorb, which we blame on ourselves. We have extensively re-written the paper to be much easier to read and accessible to a general audience. The only analyses of the data that are “idiosyncratic to songbirds” have to do with measurement of acoustic features. We have provided a simple description of how acoustic similarity was measured in the Results, and interested readers can consult the details in the Materials and methods. Our honest opinion is that the paper was poorly written overall, and that it “suffered under the weight of complexities” (as described by Reviewer #3), so that the problem was not that the analyses are idiosyncratic to songbirds. We believe the paper is no longer a difficult read, in its highly revised form. The results will be of particular interest to any readers interested in cortical-basal ganglia circuits, and to a wide audience of readers interested in learning, sensorimotor integration, and development.

[Editors’ note: the author responses to the re-review follow.]

[…] Essential revisions:This paper was difficult to read because the analyses are complicated and somewhat abstract, and many separate issues are addressed which are peripheral to the main interest of the paper; during development, tutor similarity predicted firing rates most strongly during early stages of learning, and SHELL but not CORE neurons showed decreases in response variability. This paper has thus much inherent interest but is too unwieldly as presented and how it should be revised was the subject of lively constructive discussion by the expert reviewers. In revision, the authors should focus and provide detailed responses to the points below.1) The main revision must be to simplify the message of the paper by focusing ONLY on the song similarity analysis.a) The 'premotor vs non premotor' analyses are not necessary to support the main finding, and given the relatively weak support in favor of a CORE-SHELL distinction on that issue, it should be removed.b) A large portion of the results is devoted to showing that juvenile syllables become more and more similar to tutor during early song development. Although previous studies have focused on the later stages of learning, song similarity is known to increase even in the early stages of learning (e.g. Aronov et al., 2008). The authors should cut substantially the lengthy description and analysis of song development to concentrate on the core issues; during development, tutor similarity predicted firing rates most strongly during early stages of learning, and SHELL but not CORE neurons showed decreases in response variability.c) The concentration on the relation between LMAN firing and the acoustic properties of the syllables is unwarranted. The issue here is that findings are very weak. The authors make a very problematic statistical claim about representation of multiple features but no single feature. This could be, in principle, true, but the analyses show simply a very weak effect, as opposed to synergy or gestalt representation. It is not possible to see anything interesting in the raw data. To quote one of the expert reviewers in discussion "The entire section about analysis of acoustic features, and all figures showing nothing apparent, should be removed, and replaced by something like "we found a very weak, but statistically significant representation of song features (t=… p=… see suppl.)."d) Another issue that undermines the clarity of the paper is the intermixing of results that apply both to LMAN-core and LMAN-shell neurons and results that contrast between these two areas. The authors should highlight similarities and differences between these two areas. As suggested by the first sentence in the discussion, the main finding of the paper applies to both core and shell neurons, and only minor differences are found between the singing-related firing in these two populations. This could be made more explicit throughout the results.

The biggest issue for all reviewers was the length and complexity of the paper, coupled with the fact that many of the issues addressed were peripheral to the main interest of the paper.

We apologize for the encyclopedic nature of the Results section. We had thought that it was important to fully characterize the activity patterns of shell neurons since no previous studies have made chronic recordings from this region. Clearly, in retrospect, that was a mistake. We have addressed this issue by focusing on the sections dealing with similarity of self-generated song behavior to memorized tutor song, as requested. We did this in the following ways: (1) We completely eliminated the section describing the correlation between neural activity and acoustic features of syllables; we agree with Reviewer 4 that as presented the findings are based on statistics and not on any biological demonstration, and rather than include 1-2 sentences we think it best to reserve these data for a future paper. (2) We also eliminated the entire section describing the variability of spiking activity across syllable renditions (along with the description of how we measured acoustic similarity of syllables, which is now contained only in the Materials and methods); this description has been replaced by two brief sentences in the first section of the Results, and the data are shown in Figure 2—figure supplement 1. (3) We shortened the description of the developmental increase in similarity between juvenile and tutor syllables; we continue to include the new finding that only syllabic utterances with low tutor similarity at the onset of song learning become more similar to tutor during song development; this pattern is important since the pattern of neural activity mirrors this behavioral pattern (which has not been described previously). (4) In the course of simplifying the text of the Results, we have attempted to highlight similarities and differences between core and shell in an organized way, as requested. (5) We eliminated the section on the time course of singing activity, but have included these data as a single paragraph at the end of the first section; the request to remove the ‘premotor vs non premotor analyses’ was confusing to us, in part because it did not appear in the comments of any of the three reviews; we have shortened and clarified the description to show that this result represents a strong difference between core and shell: the lack of a pre-motor response in shell neurons is essential to interpreting their role in skill learning; we think this issue is clear in the revised version. In summary, we condensed the first four sections of the previous version into one section, reducing the number of paragraphs from eight to three. This introductory section contains a brief description of the similarities in singing-related neural activity between shell and core, and of the one main difference that core neurons show a coordinated premotor increase in activity whereas shell neurons do not.

[Editors' note: further revisions were requested prior to acceptance, as described below.]

The manuscript is much improved and all comments were fully addressed.There are two remaining issues that should be resolved before this paper is published.1) Regressions of baseline-corrected firing rates against tutor similarity for each neuron revealed that cells in both CORE and SHELL exhibited either positive or negative associations between neural activity and degree of tutor similarity: approximately half of all neurons in each subregion showed positive slopes (r values > 0, increased firing rates for syllables with higher tutor similarity).We should have noticed this issue in the previous round, but this description is deficient. Saying that half increased and half decreased is like looking at random data, right? Instead, plot a histogram of all those slopes and/or r values, so readers can see the distribution on both positive and negative sides. Then do bootstrap (shuffling tutor similarity) to plot this histogram against a random distribution of similarities. This way, it would be possible to evaluate if this is a real correlation.

There is some confusion. Subsection “Neural activity in LMAN reflects similarity of self-generated syllables to tutor syllables” of the latest version of the paper read as follows:

“Regressions of baseline-corrected firing rates against tutor similarity for each neuron revealed that cells in both core and shell exhibited either positive or negative associations between neural activity and degree of tutor similarity: approximately half of all neurons in each subregion showed positive slopes (r values > 0, increased firing rates for syllables with higher tutor similarity) whereas the other half showed negative slopes (r values < 0, increased firing rates for syllables with lower tutor similarity) (Table 2).”

This sentence has nothing to do with evaluating whether these correlations are “real” (i.e. significant) or not, but is simply meant to present a description. The purpose of the sentence is to introduce the basic (and unexpected) idea that the firing rate of cells could either increase or decrease as a function of tutor similarity. We only address the issue of the statistical significance of these relationships in the following two paragraphs:

In the last paragraph of subsection “Neurons in both CORE and SHELL subregions of LMAN exhibit singing-related neural 105 activity in juvenile birds” we test whether individual neurons have significant correlations by performing repeated permutation tests (1,000 random shuffles for each cell, as described in that paragraph and in the Materials and methods), this analysis provides a reliable estimate of how many cells showed a significant relationship (which turned out to be 5.5% in core and 10.8% in shell).

In subsection “Neural activity in LMAN reflects similarity of self-generated syllables to tutor syllables” we test these correlations at the population level using the mixed-effects linear regression model recommended by our expert statistician. In our opinion, that analysis is highly appropriate, and is preferable to the strategy of performing a single randomization (bootstrap) for comparison to the actual data, as suggested by the reviewer. In addition, we note that total n’s and overall means are presented as part of Table 2, and the firing rates (response strengths) for each cell are plotted in Figure 5 as descriptive data so as to include all of the slopes (both positive and negative across all cells); we think that showing the actual data in this way instead of as a histogram of slopes or r values is also preferable. In summary, we did test for significance across the population, but did so using a mixed-effects linear regression model.

Possibly the reviewer thought we were making a claim about significance in the second paragraph of subsection “Neurons in both CORE and SHELL subregions of LMAN exhibit singing-related neural activity in juvenile birds”; although that is not true, it raises the possibility that other readers might also be confused. So one solution would simply be to re-write that sentence such that it is framed mainly in terms of describing changes in firing rates:

“Unexpectedly, this analysis revealed that firing rates of cells in both core and shell could either increase or decrease as a function of tutor similarity: approximately half of all neurons in each subregion showed increased firing rates for syllables with higher tutor similarity (positive slopes, r values > 0,) whereas the other half showed increased firing rates for syllables with lower tutor similarity (negative slopes, r values < 0) (Table 2).”

A “stronger” solution would be to simply omit that sentence altogether from that paragraph (it is not necessary there but was intended to make the text more accessible by introducing the idea of increases in firing rate encoding either higher or lower tutor similarity). Instead, that sentence, along with the rest of that paragraph and Figure 4, could be incorporated into the following paragraph. In my opinion the revision I have made is sufficient to alleviate any confusion, and seems more user-friendly in terms of providing a gradual unfolding of the data. However, I can make the “stronger” revision if the reviewer prefers.

2) Response strength did not differ between CORE and SHELL neurons for either excitation or suppression (Table 1; Mann-Whitney tests: singing-excited neurons U = 1768, p = 0.06, singing-suppressed neurons U = 459, p = 0.90).Maybe you should not say that there is no difference for singing excited neurons since 0.06 is almost significant. Instead, say that you see a borderline difference between core and shell, which did not reach significance level, for song-excited neurons, but no apparent effect in the song-suppressed neurons and then give the p values.If the statistical tests in comment #1 work out satisfactorily, then further review by the expert reviewers will not be required.

We agree with this comment and have emended the text in the Results to read as follows: “Excitatory response strength was marginally higher in SHELL neurons, whereas suppressed response strength did not differ between CORE and SHELL (Table 1; Mann-Whitney tests: singingexcited neurons U = 1768, p = 0.06, singing-suppressed neurons U = 459, p = 0.90).”